# Effect of droughts and climate change on future soil weathering rates in Sweden

Veronika Kronnäs[1], Klas Lucander[1], Giuliana Zanchi[1], Nadja Stadlinger[2], Salim Belyazid[3], Cecilia Akselsson[1]

[1] Department of Physical Geography and Ecosystem Science, Lund University, 223 62 Lund, Sweden
[2] Swedish Chemicals Agency, 172 67 Sundbyberg, Sweden
[3] Department of Physical Geography, Stockholm University, 106 91 Stockholm, Sweden

*Correspondence to*: Veronika Kronnäs (veronika.kronnas@nateko.lu.se)

**Abstract.** In a future warmer climate, extremely dry, warm summers might become more common. Soil weathering is affected by temperature and precipitation, and climate change as well as droughts can therefore affect soil chemistry and plant nutrition.
In this study, climate change and drought effects on soil weathering rates and release of Ca, Mg, K and Na were studied on seven forest sites across different climates in Sweden, using the dynamical model ForSAFE. Two climate scenarios were run, one medium severity climate change scenario from IPCC (A1B) and one scenario where a future drought period of 5 years was added, while everything else was equal to the first scenario. The model results show a large geographical variation of weathering rates for the sites, without any geographical gradient, despite the strong dependence of temperature on weathering,
and the strong gradient in temperature in Sweden. This is because soil texture and mineralogy have strong effects on weathering. The weathering rates have a pronounced seasonal dynamic. Weathering rates are low during winters and generally high, but variable, during summers, depending on soil moisture and temperature. According to the model runs, the future yearly average weathering rates will increase by 5–17 % per degree of warming. The relative increase is largest in the two southeastern sites, with low total weathering rates. At sites in southern Sweden, future weathering increase occurs throughout the
year, according to the modelling. In the north, the increase in weathering during winters is almost negligible, despite larger temperature increases than in other regions or seasons (5.9 °C increase in winter in Högbränna; the yearly average temperature increase for all sites is 3.7 °C), as the winter temperatures still will mostly be below zero. The drought scenario has the strongest effect in southern Sweden, where weathering during the later parts of the drought summers decrease to typical winter weathering rates. Soil texture and amount of gravel also influence how fast the weathering decreases during drought, and how
fast the soil rewets and regain normal weathering rates after the drought. The coarsest of the modelled soils dries out and rewets quicker than the less coarse of the modelled soils. In the north, the soils do not dry out as much as in the south despite the low precipitation, due to lower evapotranspiration, and in the northernmost site weathering is not much affected. Yearly weathering during the drought years relative to the same years in the A1B-scenario are between 78 % and 96 % for the sites. The study shows that it is crucial to take seasonal climate variations and soil texture into account when assessing the effects of a changed
climate on weathering rates and plant nutrient availability.

# 1 Introduction

In some regions of the world, the risk for plant nutrient deficiencies is high, for example in regions with low weathering rates where anthropogenic sulphate and nitrogen deposition have caused acidification and eutrophication, with leaching and increased vegetation uptake of base cations (BC: Ca, Mg, K and Na) from soils (Johnson et al., 2018). Large parts of southern Sweden are such sensitive and acidified areas with decreasing levels of base cations (Akselsson et al., 2013). Measurements of nutrient concentrations in pine and spruce needles in southern Sweden between 1985 and 1994, as well as in leaves in Europe between 1992 and 2009, have shown signs of base cation imbalances (Thelin et al., 1997; Jonard, et al., 2015).

Future societal demands for forestry products are expected to increase with the need of replacing fossil energy and materials (Böttcher et al., 2012). In the Nordic countries, especially Sweden and Finland, the forestry sector is a large and important sector, expected to both deliver increasing amounts of bio-energy and provide increasing carbon storage, counteracting carbon emissions (Hertog et al., 2022). Since clear cut forest is replanted, moderately increased harvesting does not lead to decreased forest cover, but it does lead to magnified nutrient removal from forests and decreased nutrient stores in soils as the new trees utilise the nutrients in the soils for their growth (Kaarakka et al., 2014). Increased harvest could lead to nutrient deficiencies, making the new trees more vulnerable to stressors such as insects, diseases and climate change (de Oliveira Garcia et al., 2018) and risk the forests' ability to store more carbon (Restaino et al., 2016). Another complicating factor is that forest growth has been increasing in the Nordic countries for a long time (Christiansen, 2014), because of management, elevated nitrogen deposition, increasing levels of $CO_2$ in the atmosphere and raised temperatures (Jörgensen et al., 2021), which has increased removal of base cations from soils already. Continued growth increases have been expected and are often included in carbon emission abatement calculations (Lundmark et al., 2014), but this would exacerbate the risk of nutrient deficiencies even further (Akselsson et al., 2007).

Base cations originate to a large extent from weathering of minerals in the soil. Weathering takes place on the surfaces of the mineral grains and is driven by the amount of moisture and concentrations of H+, OH-, $CO_2$ and dissolved organic carbon (DOC), while being impeded by concentrations of weathering products (for example base cations, aluminium, and dissolved silica; White and Buss, 2014). The texture of the soil is an important factor for weathering, since the smallest grain sizes (clay and silt) have a larger total surface area per weight than larger grain sizes (sand and gravel) (Brantley et al., 2008). Weathering rates increase exponentially with temperature, everything else equal (Brady and Weil, 1999). As different minerals have different chemical composition and different strengths of chemical bonds between their constitutions, weathering rates are different for different minerals and chemical weathering pathways, and the rates have been described for different conditions (temperature, chemical conditions, etc.) in laboratory experiments (White and Buss, 2014).

Increasing temperatures increase potential evapotranspiration, and soils in a warming climate might therefore become drier, even if the amount of precipitation is unchanged. Furthermore, climate change can lead to changes in precipitation amounts or in seasonality of precipitation, so that less of the precipitation falls during the warmer months. For Sweden, precipitation is projected to increase in many future scenarios, but with more precipitation falling in winter and possibly less during the

growing season (Belyazid et al., 2022), which might mean drier soils during the seasons when the trees need the water and the base cations.

When climate gets warmer, not only does the average precipitation change, but extreme events like droughts and heatwaves become more likely and more extreme (Kellomäki et al., 2008). One recent example of a climate change affected extreme event is the drought in parts of Europe, including Sweden, during the summer of 2018 (Toreti et al., 2019). Droughts might affect weathering and the nutrient availability, as well as the forest growth and tree mortality (Hartman et al., 2018).

Soil weathering rates both affect and are affected by concentrations of weathering products in soil solution (Brantley et al., 2008). The concentrations in the soil solution both affect and are affected by uptake by trees, and trees are affected by climate change and management (Hayatgheibi et al., 2021). Thus, the weathering is affected by a complicated set of interdependencies, where it is difficult to predict the directions of changes in soil water base cation concentrations, tree nutrient status and weathering rates in a future climate.

As a part of predicting the future capacity for carbon storage in the forest, as well as the sustainable amount of forestry production, it is therefore vital to know more about how weathering rates react to climate change and droughts. Measuring weathering rates *in situ* is not possible, and weathering rates measured in a lab setting differ a lot to actual rates in undisturbed soils in nature (Brantley et al., 2008). Weathering rates therefore need to be estimated using indirect methods or modelled. Some methods that have been used for estimating average weathering rates for a soil include mass balance approaches, that calculates weathering as the difference between measured other inputs and outputs of base cations to and from a catchment (Futter et al., 2012), historical methods, that estimates weathering by comparing the composition of soil in the upper, weathered soil layers, and the lower, less weathered C-horizon (Starr et al., 2014), and geochemical models such as PROFILE and ForSAFE (Belyazid et al., 2006). Using models, simple future scenarios can be analysed to estimate changes in average weathering rates and thus future risk for acidification or estimation of future sustainability of forestry for a future climate scenario. Recently, more advanced dynamic biogeochemical models have been used to get more detailed knowledge of weathering dynamics and future changes in weathering (Kronnäs et al., 2019; Gustafsson et al., 2018), also taking into account the effects of climate change on tree uptake.

This study aimed to describe how weathering release of base cations (Ca, Mg, K and Na) develops in a future with medium severity climate change scenario, and to investigate how it is further affected by five consecutive years of warm summer drought. For this we used the dynamic biogeochemical model ForSAFE (Belyazid et al., 2006; Wallman et al., 2005) to simulate weathering rates under two climate scenarios in different climate regions of Sweden. The model was applied to seven managed spruce forest sites located in seven different climate regions of Sweden. The sites have been monitored within the Swedish Throughfall Monitoring Network (SWETHRO) (Pihl Karlsson et al., 2011). The sites were modelled from 1900 to 2100 with a daily time step.

**2 Methods**

**2.1 ForSAFE**

The ForSAFE model is a dynamic process based biogeochemical model developed to study the effect of atmospheric deposition, climate change and forest management on tree growth, soil and runoff water chemistry and C and N cycling. It includes dynamic feedbacks between soil chemistry, hydrology, forest growth and organic material in the soil and consists of

integrated modified versions of four models: SAFE, a geochemical soil model (Alveteg et al., 1995; Martinsson et al., 2005), the hydrological PULSE model (Lindström and Gardelin, 1992), the PnET model of tree growth (Aber and Federer, 1992) and the DECOMP model for decomposition of soil organic matter (Wallman et al., 2006; Walse et al., 1998) (Fig. 1).

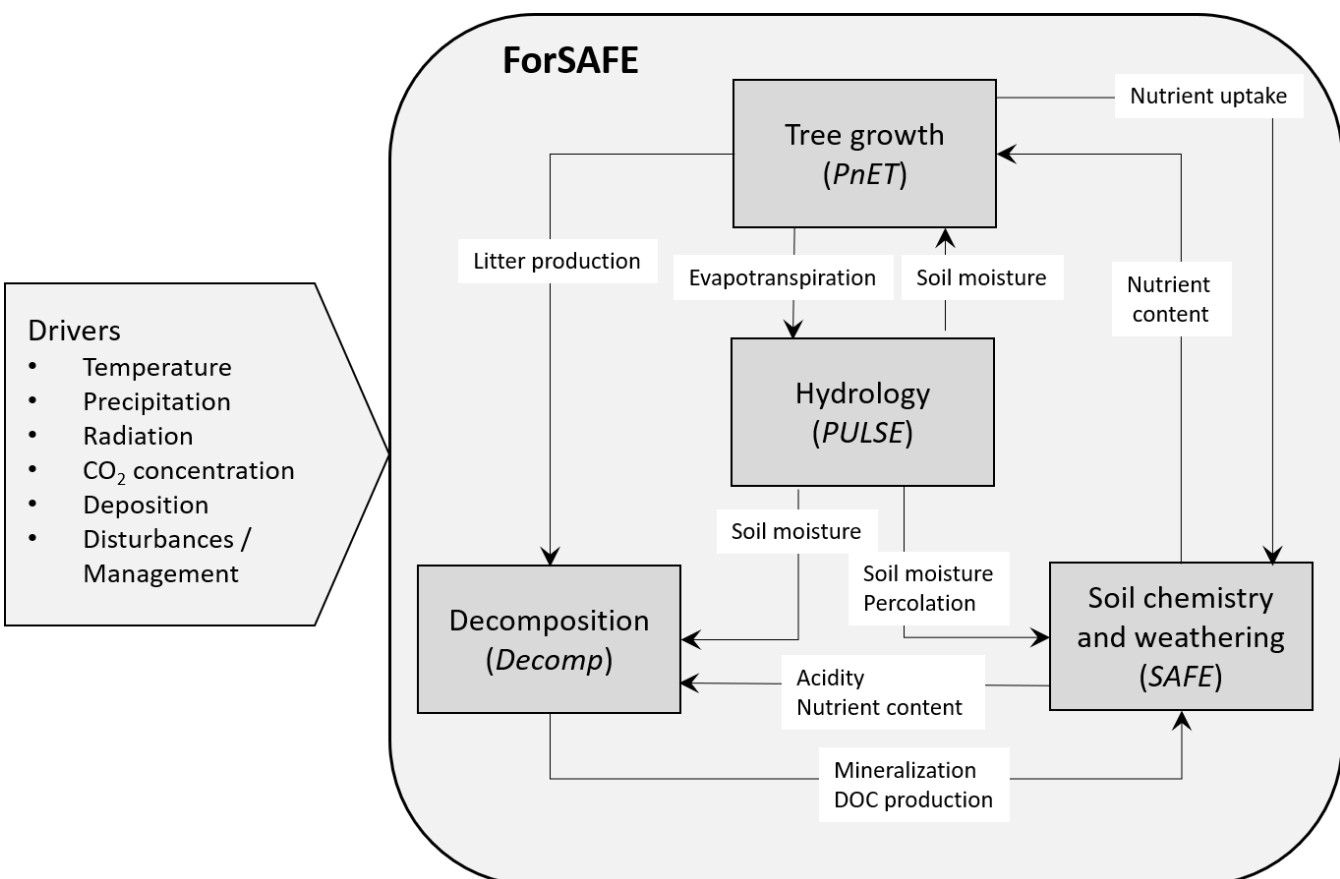

**Figure 1. Schematic illustration of the components of the ForSAFE model and the feedbacks between them. To the left, input data to the ForSAFE model are listed. From Zanchi and Brady (2019).**

The ForSAFE model is being developed in order to better answer new research objectives, such as the magnitude of weathering rates in deeper soil layers (Erlandsson Lampa et al., 2020); effects of nitrogen fertilisation in regions of different nitrogen

availability (Lucander et al., 2021); dynamics of weathering rates (Kronnäs et al., 2019); response of ground vegetation to changes in nutrient availability and acidification (Belyazid et al., 2011 and Phelan et al., 2016); changes in chemical composition of soil water from a hilltop towards a stream (Zanchi et al., 2016); effects of intensified forestry (Zanchi et al., 2021b); and phosphorous dynamics (Yu et al., 2018).

A modelled site is in ForSAFE represented by tree biomass and a soil with a discrete number of soil layers (often coinciding with 4-5 soil horizons), consisting of mineral and organic material as well as soil water. To these compartments, inputs of energy, water, base cations, carbon, nitrogen, hydrogen ions, sulphate, chloride and phosphorous are given. They are integrated by flows between them and there are also flows out of the system through downwards leaching, evaporation to the atmosphere and harvest of biomass. Weathering of soil minerals, tree growth and other chemical and biological processes occur in the compartments. Weathering in ForSAFE occurs through four chemical pathways: reactions between the mineral and water, hydrogen ions, carbon dioxide or dissolved organic acids. Total weathering in a soil layer and a time step is the sum of the weathering through all four pathways for all present minerals and depend on the chemical conditions (moisture, concentration of reactants and products) in the layer at the time step, as well as soil temperature, available minerals, and total mineral area. Soil temperature is modelled in ForSAFE, based on air temperature, snow cover, moisture, etc.. The weathering rate calculations used in ForSAFE are described further in Belyazid et al. (2022).

In this study, ForSAFE with daily time steps and one soil profile per site was used. A new subroutine with regard to hydrology, with an internal smaller time step in the calculation of the water flows, was implemented to make its handling of the infiltration during heavy rainfalls more reliable.

## 2.2 Site descriptions

Most of Sweden has a (mildly) continental climate, with southern Sweden on the border of or transitioning into a temperate climate, in the Köppen-Geiger climate classification (Kottek et al., 2006). Of the seven climate regions of this study (Fig. 2), the four regions in southern Sweden are near the border between climate classes Cfb and Dfb (warm-summer humid temperate and continental climate, both with no large seasonal differences in precipitation, four or more summer months with a monthly average temperature above 10 °C and the difference between C and D being whether the average temperature of the coldest month is above or below -3 °C). The three northern climate regions, except parts of the mountain area, are in the class Dfc (subarctic climate, same criteria as Dfb except that less than four months have a monthly average temperature above 10 °C). One site in each climate region was chosen from the SWETHRO network for this study (Fig. 2 and Table 1). Sites were chosen based on availability of necessary data for the model and dominant tree species, to make the sites more comparable across the regions. The sites are covered with productive spruce (*Picea abies*) forest and have typical Swedish conifer forest soils – thin, young, coarse tills, usually mostly made up of nutrient poor granitic and gneissic mineralogy, formed during the last deglaciation (Akselsson et al., 2007). The climate region in the Caledonian mountain region, in the northwestern part of the country, has a more varied and more base rich mineralogy than most of the country. The site in this region, Ammarnäs, is a

representative example of this, with 97 % base saturation and high soil water concentrations of especially calcium (Greiling et al., 2018).

At the sites, monthly measurements are made of deposition of acidity, $NH_4^+$, $SO_4^{2-}$, $Cl^-$, $NO_3^-$, $Ca^{2+}$, $Mg^{2+}$, $Na^+$, $K^+$, organic N and total organic carbon, using ten collectors under the canopy and (for most sites) one collector on an open field nearby. Three times per year, soil water chemistry at 50 cm depth in the mineral soil is measured using suction lysimeters. Soil properties, e.g. layer depth, soil density, texture and total chemistry of the soil, have been measured once per site in four to five soil layers (O-, A-, AB-, B- and C-horizons). In Ammarnäs, the measured total chemistry from the dug soil pit had low calcium content in the soil minerals, but very high concentrations of adsorbed calcium. The lysimeters showed that there were always high calcium concentrations in the soil water. Taken into consideration the measured amount of calcium deposition, there is no possibility that these high calcium concentrations in the soil water and adsorbed on soil particles would develop without some further source of calcium than deposition and weathering from the measured low calcium mineralogy. This indicates that there is a local variability in soil composition at the site, with either higher content of easily weatherable calcium rich minerals close to where the lysimeters are placed, or a lateral flow from a much more calcium rich soil somewhere outside of or within the site. Calcite and soils with high calcium content have been found in rock samples in the Ammarnäs area (Grimmer et al., 2016) and it is thus likely that there is a variation in calcium content and mineralogy in the soils at the site too.

Forest biomass has been measured at the sites, four times in Västra Torup and Högbränna, three times in Bordsjö, once in Ammarnäs, Hyttskogen and Södra Averstad. From these measurements, forest growth rate was calculated, as a comparison to the modelled values. The sites are productive forests, and are managed by the forest owners, following current Swedish recommendations with regard to clear cut age and thinnings. This means that they are clear cut at a younger age in southern Sweden and older in northern, where forest growth is slower. Therefor the stands were planted in different years and will be clear cut in different years. The sites Västra Torup, Södra Averstad and Holmsvattnet were clear cut recently, in 2010, 2016 and 2010, respectively.


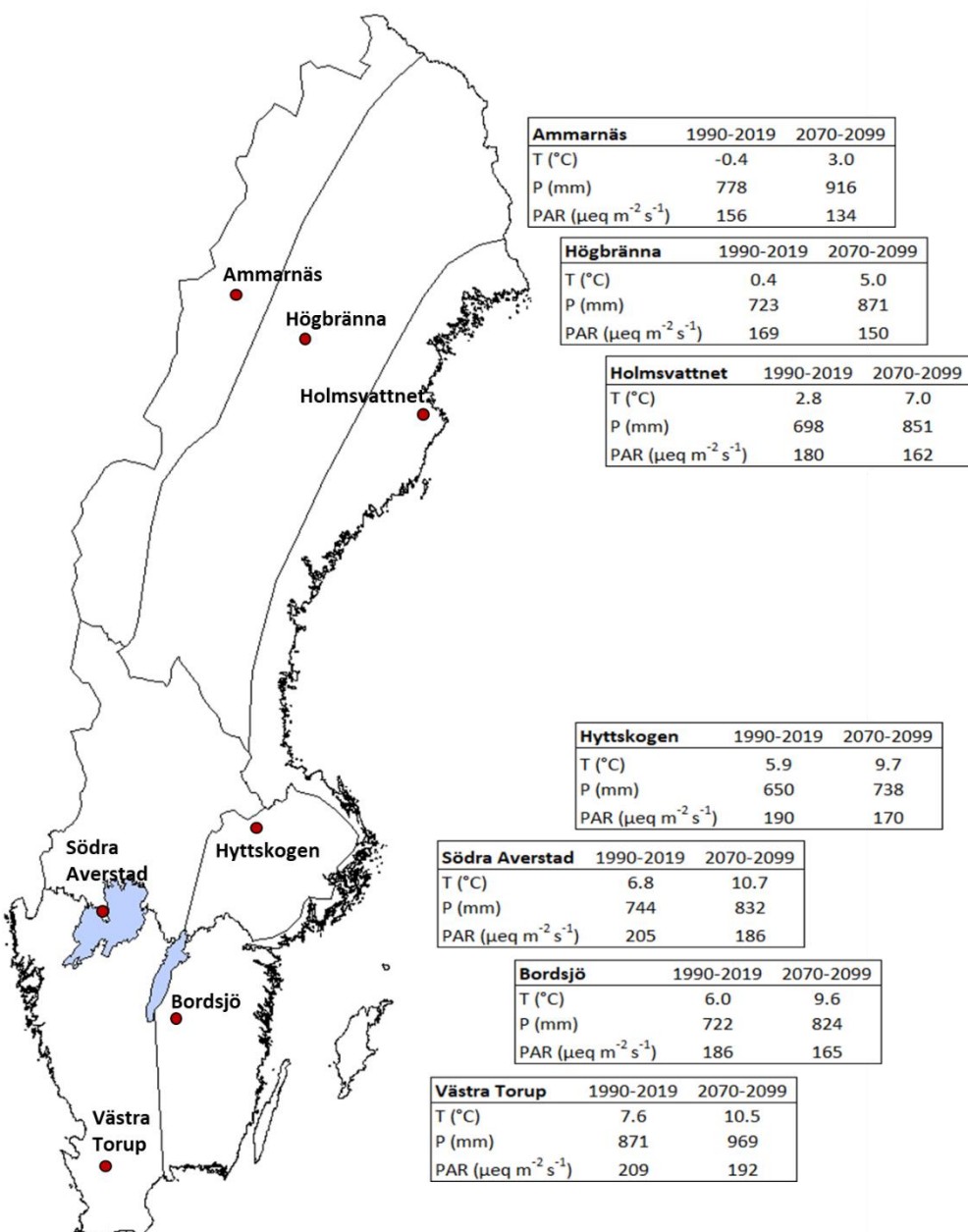

**Figure 2. The seven SWETHRO sites in their seven climatic regions, with the climatic parameters used in the modelling (T: temperature, P: yearly precipitation and PAR: photosynthetic active radiation) as arithmetic averages for the 30-year time period 1990–2019 to the left and projected climate for the A1B scenario for the time period 2070–2099 to the right.**


**Table 1. Measured data at the sites, ordered from south to north. BC stand for $Ca^{2+}+Mg^{2+}+K^{+}+Na^{+}$. ANC is acid neutralizing capacity, ANC = [BC] + [NH$_4^+$] – [SO$_4^{2-}$] – [NO$_3^-$] – [Cl$^-$]. Arithmetic averages for deposition and soil water chemistry, with standard deviation in parentheses. Base saturation and soil texture at the sites are only measured once.**

| | Västra Torup | Bordsjö | Södra Averstad | Hyttskogen | Holmsvattnet | Högbränna | Ammarnäs |
|---|---|---|---|---|---|---|---|
| **Average deposition on open field** | | | | | | | |
| Years * | 1997-2010 | 1996-2001 | 1991-2019 | 1993-2019 ** | 1992-2019 | 1996-2019 | 1992-2000 |
| Precipitation (mm) | 1025 (203) | 838 (173) | 743 (185) | 703 (112) | 607 (136) | 616 (136) | 603 (160) |
| SO$_4^{2-}$ (kg*ha$^{-1}$*yr$^{-1}$) | 6.4 (2.0) | 5.0 (1.0) | 3.8 (2.0) | 3.2 (1.1) | 2.0 (0.8) | 0.9 (0.8) | 1.3 (0.3) |
| NO$_3^-$+NH$_4^+$ (kg*ha$^{-1}$*yr$^{-1}$) | 29.1 (13.8) | 12.7 (6.9) | 7.6 (3.6) | 3.7 (2.1) | 1.7 (0.5) | 1.3 (0.5) | 3.4 (1.6) |
| Cl$^-$ (kg*ha$^{-1}$*yr$^{-1}$) | 12.0 (3.3) | 8.0 (1.6) | 6.2 (1.7) | 5.0 (1.5) | 2.1 (0.5) | 1.2 (0.5) | 1.4 (0.3) |
| BC (kg*ha$^{-1}$*yr$^{-1}$) | 24.7 (10.6) | 12.6 (5.4) | 7.3 (3.4) | 5.8 (3.4) | 2.4 (0.5) | 2.8 (0.5) | 2.9 (-) |
| **Average soil water chemistry** | | | | | | | |
| Years | 1996-2017 | 1996-2018 | 1990-2015 | 2002-2018 | 1998-2010 | 1996-2018 | 1991-2018 |
| pH | 4.6 (0.2) | 4.8 (0.2) | 4.8 (0.2) | 6.1 (0.3) | 5.2 (0.3) | 5.8 (0.2) | 6.6 (0.2) |
| ANC (µeq*l$^{-1}$) | -108 (72) / -288 (394) + | -18 (65) | 0.9 (41) | 139 (118) | 25 (23) | 56 (21) | 395 (103) |
| SO$_4^{2-}$ (µeq*l$^{-1}$) | 153 (77) | 132 (75) | 118 (53) | 85 (60) | 177 (32) | 37 (12) | 85 (18) |
| Cl$^-$ (µeq*l$^{-1}$) | 197 (99) | 172 (139) | 166 (78) | 52 (44) | 41 (18) | 22 (11) | 53 (34) |
| NO$_3^-$ (µeq*l$^{-1}$) | 0.8 (2.0) / 292 (415) + | 1.0 (1.4) | 3.3 (11.4) | 3.4 (8.3) | 3.1 (5.1) | 0.2 (0.3) | 0.1 (0) |
| BC (µeq*l$^{-1}$) | 270 (78) | 281 (167) | 288 (83) | 254 (173) | 244 (49) | 114 (26) | 532 (103) |
| Al-tot (mg*l$^{-1}$) | 1.5 (0.6) / 2.8 (3.1) + | 0.9 (0.4) | 1.0 (0.2) | 0.2 (0.1) | 0.6 (0.1) | 0.1 (0.1) | 0.1 (0) |
| DOC (mg*l$^{-1}$) | 7.3 (2.3) | 10.0 (6.0) | 11.9 (4.8) | 8.6 (3.8) | 7.0 (1.9) | 2.8 (1.0) | 12.2 (3.4) |
| **Base saturation at 50 cm** | 6.5% | 12% | 13% | 16% | 28% | 31% | 97% |
| **Distribution of soil particles sizes at 50 cm depth** | | | | | | | |
| Gravel, >2 mm (%) | 32.0 | 40.0 | 19.3 | 80.4 | 52.8 | 48.8 | 36.8 |
| Sand, 0.06-2 mm (%) | 54.4 | 51.0 | 66.5 | 16.8 | 40.5 | 47.2 | 55.2 |
| Silt, 0.002-0.06 mm (%) | 10.4 | 5.3 | 10.5 | 0.8 | 3.8 | 1.6 | 6.4 |
| Clay <0.002 mm (%) ++ | 1.6 | 1.5 | 1.9 | 0.4 | 0.8 | 0.8 | 0.8 |
| Organic matter (%) | 1.6 | 2.3 | 1.9 | 1.6 | 2.3 | 1.6 | 0.8 |

\* Deposition of BC was sometimes measured only part of this period
\*\* Data on open field deposition for Hyttskogen is from nearby sites Kvisterhult and Karsbo
 + Before and after clear cut
 ++ In Västra Torup, clay content has been measured. At the other sites, it was estimated, since clay and silt were erroneously not analysed separately (Lucander et al., 2021)

**2.3 Model input data and scenarios**

The ForSAFE model requires data on soils, forestry, deposition of ions from the atmosphere and climate. The three latter are given as time series for the time period 1900–2100, while the soil data show the state of the soil in one specific year. In this

paper, two scenarios have been modelled, with differing future climate during four of the future years, but the same forestry and deposition.

*Soil input data*

Soil layer thicknesses for four to five soil horizons, soil texture and soil chemistry (soil C, N, pH and exchangeable cations, as well as elemental composition of the soil minerals) were taken from measurements at the sites in a measurement campaign 2010–2011 (lower part of Table 1). For the mineral soil layers, mineralogy was estimated from the measured elemental composition of the soil minerals, using the A2M model (Posch and Kurz, 2007). The A2M model calculates all possible

mathematical solutions to the problem "What can the proportions of different minerals be if the elemental composition of the soil and the composition of the minerals are known?" There are more minerals than there are common elements in the soil, and therefore there exist several independent mathematical solutions with slightly different proportions, that could all theoretically be the true mineralogical composition of the sample. Any linear combination (for example the arithmetic average) of the solutions is also a possible mineralogy. For the modelling with ForSAFE, the average of the calculated mineralogies for

each soil layer was used. For the small mineralogic part of the organic layer, the mineralogy for the layer beneath it was used, as the mineral particles in the organic layer originate in the next layer beneath it. In Ammarnäs, for the modelling to be able to match the base cation content in soil water (as an average) and the base saturation, the calcite content of the soil was increased to 12 % (with quartz content reduced accordingly), since this calcite content has been seen in the Ammarnäs area (Grimmer et al., 2016) and using it in the model produced the measured soil and soil water chemistry as an average over time. The seven

sites are all coarse textured, but with Hyttskogen being very coarse textured and Västra Torup, Södra Averstad and Ammarnäs being a bit finer textured, although still with high sand and low clay content (Table 1).

*Forest management*

All sites are modelled with future clear cuts with stem only harvesting every 70 to 115 years, with increasing intervals to the north, and up to three thinnings between clear cuts, according to forestry recommendations. The exact years of harvest depend

on the historical clear cut years, which were obtained from the SWETHRO Network. The forestry scenarios used are the same as in Zanchi et al. (2021a).

*Deposition data*

The model needs time series of yearly deposition of acidifying and base ions. In the model, the yearly data are distributed to days with precipitation from the climate input data. The data, deposition of $SO_4^{2-}$, $Cl^-$, $NO_3^-$, $NH_4^+$, $Ca^{2+}$, $Mg^{2+}$, $K^+$ and $Na^+$,

was based on both measurements and modelling. The model does not differentiate between dry and wet deposition, but the dry deposition is affected by the existence or absence of a mature forest stand (Staelens et al., 2008). Therefore, both wet and dry deposition as well as clear cut years are needed to compute the deposition. In SWETHRO, open field deposition and throughfall deposition in the forest stands are measured monthly at each site. On some SWETHRO sites across Sweden, the relation between wet- and dry deposition for the more biologically active substances ($NO_3^-$, $NH_4^+$, $Ca^{2+}$, $Mg^{2+}$, $K^+$) are  measured on

surrogate surfaces under roofs (Karlsson et al., 2019). From these measurements, wet- and dry deposition at the sites were calculated for all years with measurements (Table 1). Open field deposition consists of mainly wet deposition and a small

percentage of dry deposition on the measurement equipment (Grennfelt et al., 1985; Persson et al., 2004; Granat, 1988), so wet deposition of all modelled ions was calculated from the open field deposition. $SO_4^{2-}$, $Na^+$ and $Cl^-$ were assumed to not interact significantly with the foliage of the trees, which means that the throughfall measurements of these ions were equal to

total deposition. Dry deposition of $NO_3^-$, $NH_4^+$, $Ca^{2+}$, $Mg^{2+}$ and $K^+$ were calculated using data from SWETHROs surrogate surface measurements (Karlsson et al., 2019) and throughfall of $Na^+$ at the site.

For all modelled years, except when measurements were available at the sites, wet deposition of base cations and chloride were kept constant at the average of the calculated wet deposition from above. For $SO_4^{2-}$, $NO_3^-$ and $NH_4^+$ deposition scenarios between 1900 and 2100 were obtained from the CLEO program (Climate Change and Environmental Objectives;

Naturvårdsverket, 2016) and from the ECLAIRE program (Effects of Climate Change on Air Pollution and Response Strategies for European Ecosystems; ECLAIRE 2021) and were downscaled to the sites using the measured wet and dry deposition. The deposition scenarios were developed using the same GCM model and climate scenario (A1B) as the climate data used. Data from ECLAIRE were used from 1900 to 1960 and from CLEO from 1961 to 2100, as neither of the sets covered the entire time period.

To simulate the effect on dry deposition of having no or smaller trees after clear cutting, dry deposition was lowered to zero at clear cut and was progressively increased for 30 years back to what it would have been without clear cutting.

*Climate data*

The climate parameters needed for the ForSAFE modelling are time series of daily average, minimum and maximum temperature, daily precipitation, daily average photosynthetic active radiation (PAR) and daily average $CO_2$ concentration of

the atmosphere, from year 1900 to 2100. The future parts of the time series are the climate scenarios. In this paper we use two climate scenarios: a scenario with climate based on the IPCC scenario A1B (Nakićenović et al., 2000) and a drought scenario where the climate parameters for five consecutive years have been changed to years with very warm and dry summers, but all other years follow the A1B scenario. Five consecutive years of dry summers were used to allow time for cumulative effects of the drought.

The climate parameters, except monthly precipitation, are not measured at the SWETHRO sites. Instead, measured temperature data from nearby climate stations from SMHI for the period 1981–2010 were used to bias-correct modelled climate data from the regional climate model RCA3 (Kjellström et al., 2005), based on the global climate model ECHAM5-r3 (Roeckner et al., 2006). The daily modelled climate data from RCA3, average temperature and precipitation, covered the years 1961–2099, using the A1B scenario for the future years (see below). Temperature data was bias-corrected using methods in Hempel et al.

(2013; algorithms 25–26), to obtain measured distributions of temperatures, as the modelled data had a smaller spread between minimum and maximum values than the measured data during the time period where both time series were available. For the time period 1900-1960, which was not covered by the RCA3 data used, we constructed data by randomly assigning each year with one of the years 1961–1970. This time period is needed for the modelling of the tree stands present at the sites today, but the exact climate is of less importance than during the time periods used for comparison of weathering rates. For the year 2100,

the same values were assumed as for 2099.

Daily values of PAR from 1960 to 2100 were retrieved from ECLAIRE. The data consisted of short-wave radiation for the years 1960–2100 for Europe and were developed by the Rossby Centre at SMHI 2012/2013 (pers. comm. Magnuz Engardt). The data was converted from $W/m^2$ into PPFD (Photosynthetic Photon Flux Density) and reduced to only contain the PAR spectra (400–700 nm) (see further Montieth and Unsworth 2008 in Klingberg et al., 2011). For the years 1900–1959, PAR

data from 1961 to 1970 was assigned in the same way as was done for the temperature and precipitation. Monthly averages of these data were lower than the monthly data used in previous ForSAFE modelling (e.g. Kronnäs et al., 2019), but comparison with modelled PAR data from SMHI (Landelius et al., 2001), and measured PAR data from the ICOS network (Carrara et al., 2018) showed that the ECLAIRE data were on the right level and that the older data (which were downscaled from a global model in a simpler way) had most likely not been adjusted enough for the effect of cloudiness.

For daily $CO_2$-values, the same time series were used for all sites. The yearly trend was taken from the A1B scenario and daily values were constructed with a schematic within year variation, to obtain lower than yearly average $CO_2$ concentration during the growing season.

*Scenario A1B*

The IPCC climate scenario A1B (Nakićenović et al., 2000) with a regional down scaling (Simpson et al., 2012) was used as

the A1B scenario. In this medium severe climate change scenario, in line with current emissions (Schwalm et al., 2020) yet far more severe than what is accepted in the Paris agreement (United Nations, 2015), the $CO_2$ concentration in the atmosphere increases to about 720 ppm in the year 2100. In Sweden, this increase leads to a warmer and wetter climate with slightly lower incoming radiation, as averages over 30 years (Fig. 2). The changes are not uniform throughout the year: both increase in temperature and precipitation and decrease in PAR values are largest during winter months. In part of southern Sweden,

monthly precipitation is projected to decrease slightly during some summer months. The larger increase in precipitation during winter than during summer together with the warmer temperatures might, in some regions, lead to drier conditions during summertime.

In the Köppen-Geiger climate classification, in 2070–2100 according to the A1B scenario, the four southern sites will be well into the Cfb class and even the northern coastal site will have transitioned into it, as their monthly average temperature of the

coldest month of the year will no longer be below -3 °C according to the A1B scenario. The northern inland site will have shifted into Dfb, as it will have five months per year with average temperature above 10 °C, and only the northernmost site will still be in the Dfc class, with only three months per year with an average temperature above 10 °C.

*Drought scenario*

In addition to the A1B scenario, a scenario with strong drought events was simulated. This scenario was identical to the A1B

scenario except for five consecutive years of strong summer drought events inspired by the very dry summer of 2018 (described e.g. in Toreti et al., 2019), when several months had higher than normal temperatures and lower than normal precipitation, leading to very low SPEI-values (SPEI: Standardized precipitation evapotranspiration index, Beguería et al., 2014) over an unusually large area, covering several countries in northern Europe, including Sweden. In the drought scenario of this paper, five consecutive years with higher temperatures and lower precipitation during summers were used. This simulates an

unusually long period of very dry summers, to simulate cumulative effects of several very dry summers. At some of the sites, the years 1974-1976 were approximately as dry, in SPEI values, as the simulated drought, but for a shorter time and without being unusually warm at the same time, and other sites had not had multiyear severe summer drought periods according to measurements starting in 1900 (Beguería et al., 2014). The drought scenario was modelled because extreme events, both wet and dry, are projected to become more common and more severe in the future, and can have large consequences on the ecosystems. The simulated dry years occurred in the second half of the 21$^{st}$ century, at a time when the forests were mature (to make the scenario more comparable between sites). Due to differing planting and clear cutting years for different sites and thus different periods with mature forest, the extreme drought years could not be the same years for all seven sites. The drought was modelled to occur in 2070–2074 in Västra Torup and Södra Averstad, and in 2090–2094 for the other sites. During April to July of these dry years, temperature and PAR were increased and precipitation was decreased according to Table 2, compared to the averages for each site of the previous ten years. August to March of the five drought scenario years had the average climate of the respective month of the ten previous years, since precipitation, temperature and PAR-values were normal in 2018 during those months. The numbers in Table 2 were based on the difference between 2018 and 2008–2017 in SMHI:s measuring sites at Osby and Växjö in southern Sweden (SMHI, 2021) and Fig. 3 shows the resulting temperatures, precipitation and PAR-values for the sites. The drought scenario results in consistently warmer, drier and sunnier weather in the months April to July compared to the same months in the A1B-scenario, even though the drought scenario is based on the weather of the preceding decade and is independent of the weather during the actual years of the drought scenario (since it is built on the weather in 2018, which for obvious reasons could not be compared to a year 2018 without drought).

**Table 2. Increase in temperature and PAR and decrease in precipitation (P) during the dry years of the drought scenario, compared to the ten previous years of the A1B-scenario.**

|  | ΔT (°C) | P drought/P base | PAR drought/PAR base |
|---|---|---|---|
| **Jan-Mar** | 0.0 | 100 % | 100 % |
| **Apr** | 1.6 | 72 % | 94 % |
| **May** | 4.2 | 25 % | 121 % |
| **June** | 3.0 | 39 % | 114 % |
| **July** | 3.5 | 26 % | 128 % |
| **Aug-Dec** | 0.0 | 100 % | 100 % |

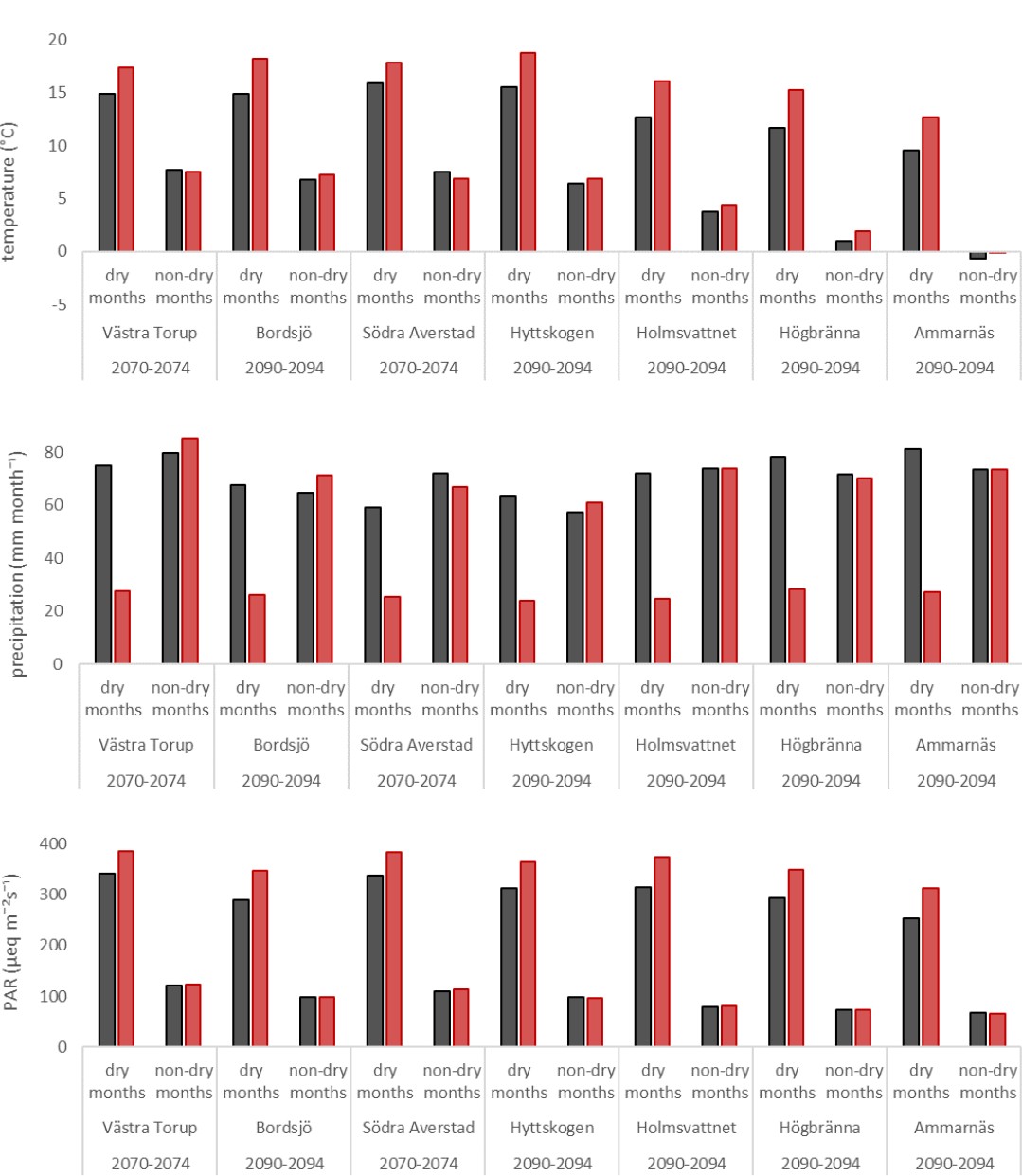

**Figure 3: Monthly average temperature, precipitation and PAR (photosynthetic active radiation) during the A1B-scenario (black) and the extreme drought scenario (red) for months with drought (April to July) and months without drought (August to March). For Södra Averstad and Västra Torup, the drought occurs during the five years 2070–2074 and for the other sites it occurs during 2090–2094. The sites are ordered from south to north in the figure, with the southernmost (Västra Torup) at the left and the northernmost (Ammarnäs) at the right.**

**3 Results**

**3.1 Weathering rates 1990–2100, A1B-scenario**

Weathering at all sites is very dependent on season (Fig. 4 and 5), with lower and less variable winter weathering and higher and more variable summer weathering. Daily weathering rates averaged over a 30 year period (Fig. 4) show a broader, more cut off summer peak of weathering in southern Sweden, and a higher, narrower peak for the northern sites. This difference is explained by longer warm periods and greater inter-annual variability of summer weathering in southern Sweden, where relatively dry periods during summers are common. At the northern sites, on the other hand, soil moisture is almost always adequate and soil temperature is thus determining for the weathering. Yearly average soil temperature at 50 cm depth at the sites in this study, is warmer than yearly average air temperature by 2 ℃ (0.6–4.1 ℃) in 1990–2019, but this difference decreases to 0.8 ℃ (0.1–2.4 ℃) in 2070–2099, due to shorter periods of snow cover. Since the difference between air and soil temperature is largest during cold winter temperatures when weathering rates are very low, the diminishing difference between air and soil temperature has a negligible effect on weathering.

Weathering also strongly depends on mineralogy and texture of the site. Therefore there is no clear north to south pattern in the amount of weathering within the considered group of sites, despite the strong temperature gradient between the sites (Fig. 2). For instance, the site in the coldest climate, Ammarnäs, with a favourable mineralogy, has more than ten times as high weathering as the southern site Bordsjö, with coarse texture and less of the easily weatherable minerals.

Future weathering increase, according to the simulations, at all sites for all seasons from the 30-year time period 1990–2019 to 2030–2059 and further to 2070–2099 (Fig. 4 and 5). Depending on future climate change and the size of the trees in each 30-year period (large trees use much more water than newly planted trees and thus affect soil moisture levels), the difference between the three time periods will vary with site according to the model. At Bordsjö, for example, the forest will be clear cut in 2030 in the future forestry scenario, which means that the entire 30-year period of 2030-2059 will have no trees or small trees, leading to a higher amount of soil water and an enhanced weathering, compared to the situation in 2070-2099 when the trees are mature. This leads to almost no increase in weathering between the two later time periods, even though the temperature is still increasing. Holmsvattnet, on the other hand, is clear cut in 2010, which means that the first of the three time periods (1990-2019) is partly affected by a clear cut yielding increased weathering rates, which means that there will be only a small increase in weathering between 1990-2019 and 2030-2059. During the last of the 30-year period, precipitation at Holmsvattnet has increased by a 100 mm per year compared to 2030-2059, giving a large increase in weathering amount between 2030–2059 and 2070–2099 (Fig. 4).

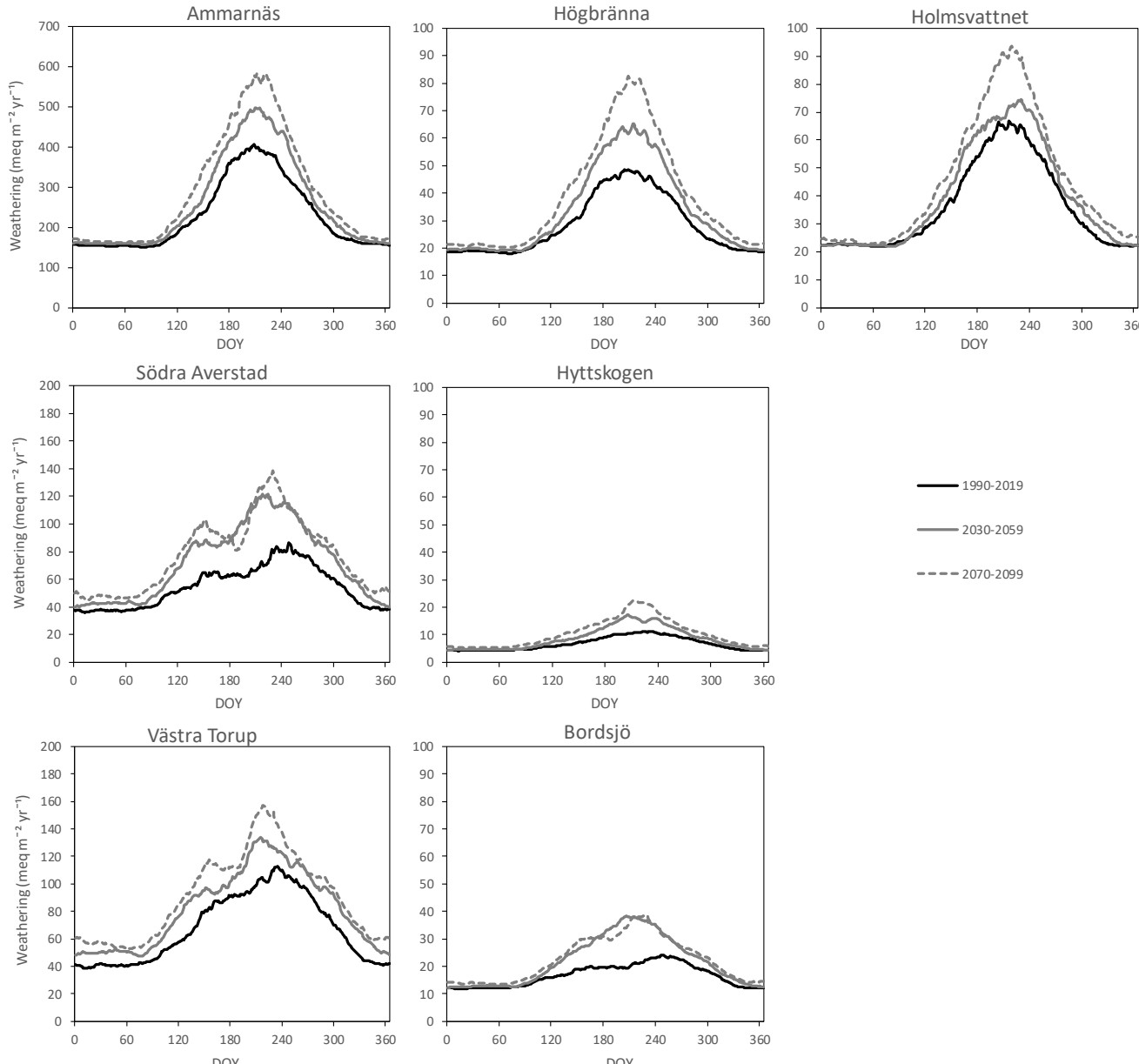

**Figure 4. Weathering variation over the year (day of the year on the x-axis), average for three different 30-year periods in the A1B scenario. Weathering rates are the sum of Ca, Mg, K and Na for the humus layer and the upper 50 cm of mineral soil. Note that the sites in the left-hand column have different y-axis scales than the other four sites. The upper three sites Ammarnäs, Högbränna and Holmsvattnet are situated in northern Sweden and the sites in the middle and lower part of the figure lie in southern Sweden.**

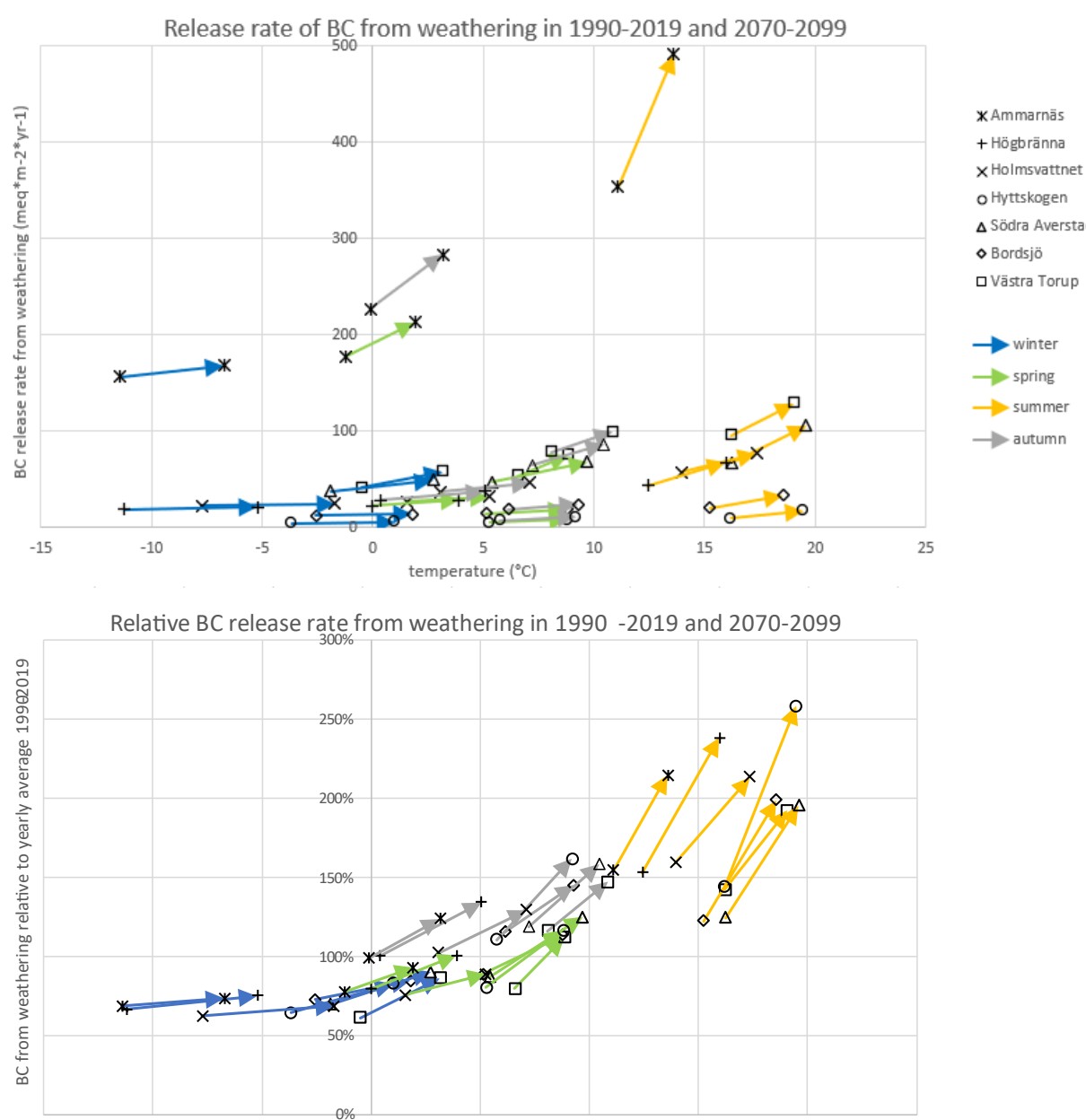

**Figure 5.** Increase between the two time periods 1990-2019 and 2070-2099 in average seasonal weathering of Ca+Mg+K+Na (total numbers above and relative to yearly average 1990-2019 below), with temperature on the x-axis, for different seasons and sites. The seasons are three calendar months long each: winter is Dec.-Feb., spring is March-May, summer is June-Aug. and autumn is Sept.-Nov. The sites are ordered from north to south in the legend, with the northernmost site Ammarnäs at the top.

The future average yearly weathering at the sites will increase between 2 % and 8 % per decade, or 5 % to 17 % per centigrade of average yearly warming, according to the model. By season, the largest future increase in weathering will generally occur in the summer months. In southern Sweden, the increase in weathering during spring and autumn will also be large, and even during winter there will be an increase in weathering of 6 % – 8 % between the time periods 1990–2019 and 2070–2099. In dry and coarse textured Hyttskogen, weathering increase during summer is calculated to 26 %, due to less dry conditions in the future than today. In northern Sweden, temperature increase during winter is projected to be larger than during any other season at any site, but weathering increases are still projected to be very small, since temperatures will still be well below zero for most of the season. The northern sites will also normally have sufficiently high soil moisture during summer to not inhibit weathering. In absolute values (Fig. 5, upper part), Ammarnäs will by far have the largest increases in weathering. In relative numbers (Fig. 5, lower part), Hyttskogen will have the largest increases in weathering, as it is so drought limited today, and will be slightly less so in the future, according to this downscaling of this climate scenario.

## 3.2 Effect of droughts

In the drought scenario, the added precipitation during May to July is much lower and the temperature higher than in the A1B scenario (Fig. 3). In the modelling, this leads to drastically decreasing levels of soil moisture during the summer (Fig. 6), with a much lower summer average than in the base scenario (Table 3) and lower weathering rates at all sites (Fig. 7, Table 3). In the southern sites, soil moisture is relatively close to the wilting point during large parts of the summer, whereas in the northern sites, soil moisture does not decrease as fast, due to lower water demand by the trees, and thus does not reach as low levels. In this drought scenario, precipitation returns to normal in August, which causes weathering rates to increase towards normal levels in late summer, autumn and winter. When normal precipitation resumes in August, soil texture influence how fast the soil rewet (Fig. 6). Soil moisture reaches field capacity at 50 cm depth in just three weeks in the site with very coarse texture, Hyttskogen, and weathering thus increases quickly to normal summer values for this site. In the less coarse soil of Västra Torup, soil moisture increases more slowly, and when field capacity is reached after two months, autumn temperatures have set in. This means that weathering is affected faster and stronger by the drought and the rewetting in the coarsest soils. The weathering during the drought years is 2 % – 22 % lower for individual base cations than under the A1B scenario.

There are some individual years on some sites when weathering in the A1B scenario is as low as in the drought scenario, because of low precipitation in this scenario as well, or because of a relatively cool summer. All the southern sites have at least two years with some period of dry conditions during summer already in the A1B scenario, but usually not as severe as in the drought scenario. For instance, Hyttskogen has four dry summers in the A1B scenario. At the northern sites, years with low summer weathering in the A1B scenario are instead years with relatively cold summers. The corresponding years in the drought scenario have much warmer temperatures, which in these northern sites compensate for the drier conditions.

The effect of the drought is slightly smaller on magnesium release from weathering and slightly larger on potassium and sodium release from weathering, but these differences vary depending on what minerals are present in the soils (Table 3).

**Table 3. Comparison of soil moisture and weathering in the A1B scenario and in the drought scenario. Average values with standard deviation in parentheses. The time period is the five years when the drought scenario differs from the A1B scenario – 2070–2074 for Västra Torup and Södra Averstad and 2090–2094 for the other sites. The sites are ordered from the south to the north in the table.**

| | V. Torup | Bordsjö | S. Averstad | Hyttskogen | Holmsvattnet | Högbränna | Ammarnäs |
|---|---|---|---|---|---|---|---|
| **Number of dry summers** | | | | | | | |
| A1B scenario | 2 | 2 | 3 | 4 | 0 | 0 | 0 |
| Drought scenario | **5** | **5** | **5** | **5** | **5** | **5** | **5** |
| **Average (June, July, August) soil moisture (%)** | | | | | | | |
| A1B scenario | **14.6 (4.0)** | **12.2 (5.0)** | **12.8 (3.7)** | **9.1 (1.4)** | **14.0 (2.0)** | **16.0 (1.6)** | **20.1 (1.6)** |
| Drought scenario | 8.9 (2.7) | 6.4 (2.4) | 8.3 (3.1) | 7.9 (0.7) | 8.5 (1.9) | 10.7 (2.1) | 14.7 (2.5) |
| **Average yearly weathering (meq m$^{-2}$ yr$^{-1}$)** | | | | | | | |
| Ca A1B scenario | 19.4 (9.2) | 4.5 (2.4) | 17.6 (8.6) | 2.7 (1.7) | 12.3 (7.5) | 7.9 (5.0) | 272.3 (140.2) |
| Ca drought scenario | 16.9 (6.0) | 4.2 (1.8) | 13.7 (6.2) | 2.5 (1.4) | 10.6 (5.1) | 6.6 (3.4) | 262.9 (123.4) |
| Ca drought/A1B | 87% | 93% | 78% | 92% | 86% | 84% | 97% |
| Mg A1B scenario | 5.2 (2.5) | 1.3 (0.6) | 3.7 (1.6) | 1.2 (0.7) | 10.0 (5.8) | 4.7 (2.7) | 4.0 (2.3) |
| Mg drought scenario | 4.5 (1.6) | 1.2 (0.5) | 2.9 (1.1) | 1.1 (0.6) | 8.8 (4.0) | 4.0 (1.9) | 4.0 (2.1) |
| Mg drought/A1B | 87% | 94% | 80% | 93% | 88% | 85% | 98% |
| K A1B scenario | 17.4 (7.3) | 4.9 (2.2) | 13.6 (5.3) | 2.9 (1.5) | 4.9 (2.4) | 7.1 (3.6) | 2.6 (1.3) |
| K drought scenario | 14.7 (5.1) | 4.5 (1.5) | 10.6 (3.9) | 2.6 (1.1) | 4.2 (1.5) | 5.9 (2.5) | 2.5 (1.1) |
| K drought/A1B | 85% | 92% | 78% | 91% | 85% | 84% | 96% |
| Na A1B scenario | 41.5 (17.2) | 11.0 (4.7) | 38.0 (14.7) | 3.2 (1.6) | 17.7 (8.8) | 19.5 (10.0) | 14.3 (6.3) |
| Na drought scenario | 35.1 (11.8) | 10.2 (3.4) | 29.5 (10.1) | 3.0 (1.2) | 15.1 (5.4) | 16.5 (6.9) | 13.6 (5.4) |
| Na drought/A1B | 85% | 93% | 78% | 92% | 85% | 85% | 95% |
| BC A1B scenario | 83.5 (36.0) | 21.6 (9.9) | 72.8 (30.1) | 10.0 (5.5) | 45.0 (24.4) | 39.2 (21.2) | 293.3 (150.1) |
| BC drought scenario | 71.3 (24.2) | 20.1 (7.1) | 56.6 (21.1) | 9.2 (4.3) | 38.8 (15.7) | 33.1 (14.7) | 283.0 (132.0) |
| BC drought/A1B | 85% | 93% | 78% | 92% | 86% | 84% | 96% |

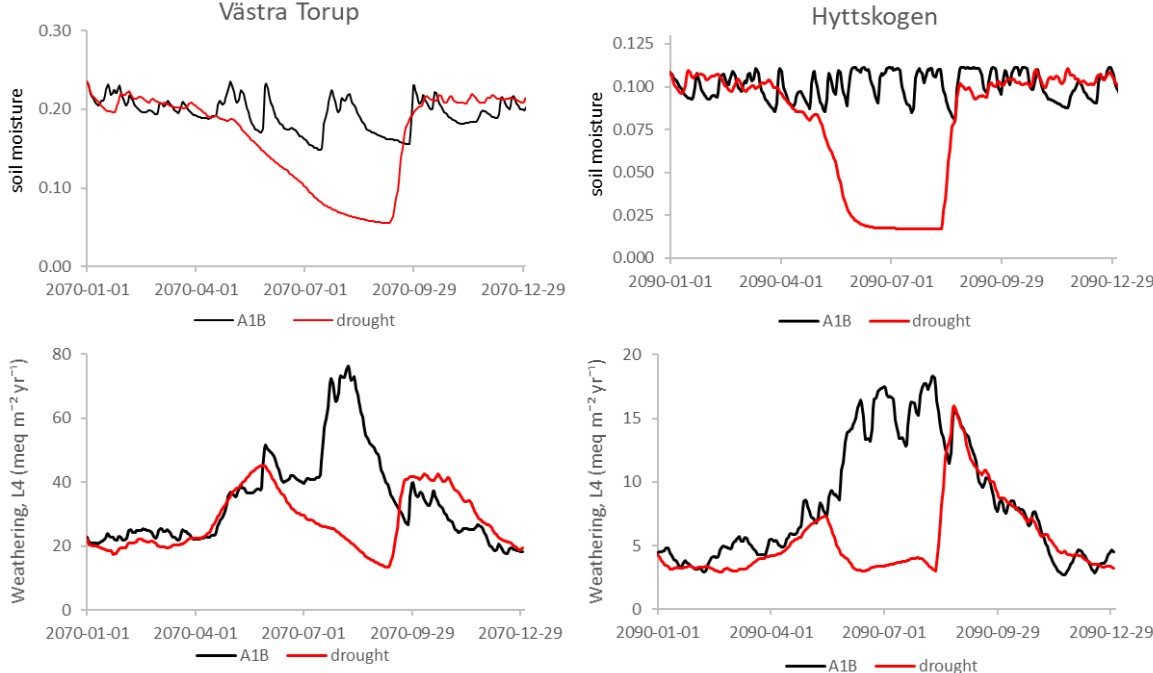

**Figure 6. Different reactions of soil moisture (above) in sites with different texture, during the first of the drought years: Västra Torup is less coarse textured than Hyttskogen is, and retains water for longer, but also infiltrates water slower when precipitation resume. This also affects the reaction of weathering rates of base cations to droughts (below).**

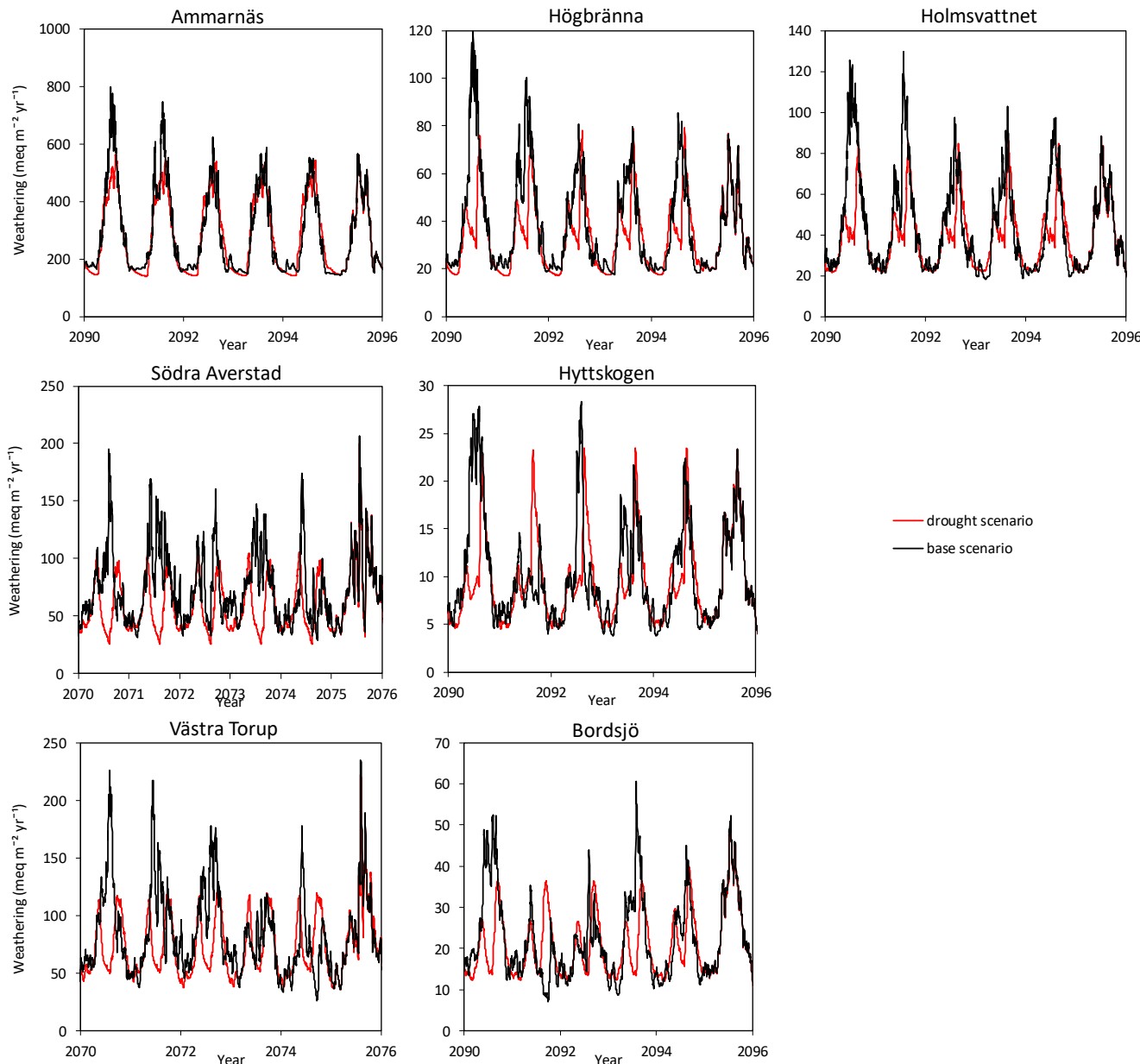

**Figure 7. Weathering under the drought scenario (red) compared to the A1B scenario (black). The first year after the drought is also included. Weathering rates are the sum of Ca, Mg, K and Na for the humus layer and the upper 50 cm of mineral soil. For Södra Averstad and Västra Torup, the drought occurs during the five years 2070–2074 and for the other sites it occurs during 2090–2094. The upper three sites Ammarnäs, Högbränna and Holmsvattnet are situated in northern Sweden and the sites in the middle and lower part of the figure lie in southern Sweden.**

## 4 Discussion

### 4.1 Weathering rates in a future climate, A1B-scenario

This study clearly shows increasing weathering rates under future climate change in Sweden. The largest modelled future increase in weathering will generally occur during the summer months when weathering is largest. According to the modelling, summer weathering will consistently increase in the north, where water availability will not decrease, whereas in the south summer weathering will depend on the combined effect of higher temperatures and change of soil moisture. The effect of warmer winters on weathering will be largest in the south, where above zero winter temperatures will become more and more common. Despite large increases in winter temperatures in the north, winter temperatures there will still usually be well below zero and thus weathering will still be low. The different seasonal distribution of the weathering increase has implications for plant nutrition. In southern Sweden, part of the weathering increase due to climate change occurs in late autumn, winter and early spring time, when plants are inactive, whereas in northern Sweden, there is almost no weathering increase in winter and little in spring and autumn. At the same time, with climate change, the growing season lengthens dramatically (Jin et al., 2019), and thus precipitation and weathering during spring, autumn and in southern Sweden even winter, will have an increasing significance for the ecosystem, especially in areas with high occurrence of summer droughts. This highlights the importance of using dynamic models with high resolution temporal climate data to assess effects of climate change on weathering rates and plant nutrition. The sites in this study are not growth limited by base cations (as very few Swedish forests are), but seasonal imbalances of availability and needs of base cations could still exist. A next step could be to study the exact seasonal dynamic of weathering, leaching and uptake of nutrients, to see if short periods of problematically low concentrations of base cations occur during part of the year at any site. For this, more detailed measurements of the timing of uptake to trees might be needed, since this is not available for the sites in this study. On a long-term basis, base cation deposition, net uptake to trees, leaching and weathering are of comparable sizes (around 4-60 meq m$^{-2}$ year$^{-1}$), except weathering and leaching in Ammarnäs, which are higher. At the southern sites, deposition is higher than weathering, and the opposite is true at the northern sites.

In the southern sites, except Västra Torup, the soil water is successively acidifying during the 21$^{st}$ century, according to the model (Fig. A.1 in Appendix A). The future growth of the forest is faster than the growth today, in this downscaling of the A1B-scenario and this ForSAFE-modelling. The inputs of base cations to the ecosystem through weathering and deposition are not large enough to fulfil the needs of the vegetation without depleting the stores in the soils and increasing the future risk of nutrient deficiencies, despite of the increased weathering rates. Such sites are common in southern Sweden according to Akselsson et al. (2016). Soil acidification itself also affects weathering rates, increasing it or decreasing it (thereby decreasing or increasing the soil acidification), depending on the changing pH, dissolved organic carbon and concentrations of dissolved aluminium, where aluminium inhibits weathering and the other two increase it (Kronnäs et al., 2019).

According to this study, there is no decrease of soil moisture due to climate change in the future for these sites. The four sites in the south experience summer droughts and limitations of tree growth because of water stress already today, which, according to Ruiz-Pérez and Vico (2020) is common in southern Scandinavia. The model results indicate no increased growth limitation

due to water stress in the future, as plants become more water efficient at higher concentrations of atmospheric $CO_2$ (Toreti et
al., 2020) and this compensates for the effect of higher temperature of the A1B scenario summers. Another ForSAFE modelling
study in Sweden, where the more extreme A2 scenario was used, show increasing water stress in southern Sweden in the future
(Belyazid and Zanchi, 2019). In Cheng et al. (2017), most models show decreasing soil moisture in southern Sweden for the
future scenario RCP4.5 (less severe than the A1b scenario) and for all of Sweden in the very severe RCP8.5 scenario. If soil
moisture decreases in southern Sweden, as these studies indicate, the projected increase in weathering might be an
overestimation and risk for nutrient deficiencies and acidification of the soils in southern Sweden would be even higher than
the result in this paper indicate.

The long term average weathering from this study compared well with previous studies of weathering in Swedish forest soils.
Five of the sites in this study have previously been modelled with the steady state model PROFILE (Kronnäs et al., 2019;
Akselsson et al., 2021). The future long term average weathering increase per degree of temperature increase from the present
study, 6 % °C$^{-1}$ as an average for all sites, is close to the value of 7 % °C$^{-1}$ in a ForSAFE modelling of Västra Torup in Kronnäs
et al. (2019). A larger Swedish ForSAFE study (Belyazid et al., 2022) with over 500 sites, found that the average BC
weathering increase per degree of temperature increase was 6.7 % °C$^{-1}$. A similar average weathering increase per degree of
warming was found in a Canadian study (Houle et al., 2020), in which a small decrease in soil water content and an increase
in BC weathering (around 7 % °C$^{-1}$ as an average for the sites in the study) was found. With the exception of Kronnäs et al.,
(2019), none of these studies have investigated seasonal dynamics in weathering.

## 4.2 Effect of extreme drought event

The severe drought leads to a lower weathering than the A1B scenario in all climate zones. This happens despite the southern
sites having summer drought to some extent already in the A1B scenario (as seen on the flattened shape of the curve in Fig. 4,
where the highest weathering rates do not occur in the midst of summer), and the northern sites not drying out as much during
the drought as the southern sites, because of lower evapotranspiration.

The lower weathering increases the risk of BC depletion and nutrient imbalances in trees, which in its turn can decrease their
capacity to cope with droughts (Hartmann et al., 2018).

The texture of the soils influences how they respond to drought and precipitation. Coarser soils have lower field capacity and
wilting points, which means that they hold less water in both normal and dry conditions and have lower weathering rates (both
because of their lower moisture levels and their lower exposed surface areas). They also respond quicker to changes: they dry
out quicker, but precipitation also infiltrate to deeper soil layers faster. In this drought scenario, when normal precipitation
resumes in August, the site with the coarsest texture, Hyttskogen, has a short period of normal summer level of weathering
before autumn temperatures set in. In the more fine-grained site Västra Torup on the other hand, weathering rates are normal
in early summer, despite lack of precipitation, as the soil holds water better, but do not increase as fast in August as in
Hyttskogen. Since the extreme drought scenario in this study has normal levels of precipitation during September to March,

every growing season of the drought event starts with normal levels of soil water according to the model. A future extreme drought event might have low precipitation during all months over several years and thus have an even larger effect on weathering rates, vegetation and long-term soil water and ground water levels, but in this study we chose to base our scenario on an existing event that has been well described.

### 4.3 Model and measurement limitations and development

Measured and modelled ANC (acid neutralizing capacity) in the soil water for the seven sites are shown and discussed in Appendix A. The model and measurements agree better on average chemistry than on the temporal variation. Discrepancies between measured and simulated water chemistry can be due both to limitations in the model and in the measurements (which are only taken three times per year and thus do not capture the temporal variation in detail). The measured temporal variations in soil water concentrations are larger than in the model results for most sites, both for ANC and most modelled ions (Appendix A; Zanchi et al., 2021a). This indicates that the variation in soil water chemistry is underestimated in the model, which may be due to the model having one set of concentrations for each soil layer, no horizontal spatial variation, and no variation in direction of horizontal water flows. It might also indicate that the variation in weathering rates might also be underestimated. Measurements of different variables at the same site, during roughly the same time period, can sometimes give concentrations and flows that do not add up, because of the variability in nature, both in space and time. For instance, in Ammarnäs, the measurements of total chemistry of the soil layers, taken from one soil pit, gave a mineralogy that could not by themselves lead to the measured soil water chemistry in the lysimeters, because of low weathering capacity. There might be different mineralogies in different parts of the site or there might be water influx from an area with more easily weathered mineralogy outside of the site. Other studies confirm that the region has a larger variation in mineralogy than usual for Sweden (Grimmer et al., 2016; Greiling et al., 2018). Measurements of soil properties in more locations within and around the site, as well as mapping of soil water flows, would be needed to be able to understand how the site functions and how large the weathering rates are in different parts of the site.

Södra Averstad also shows the importance of the soil moisture on the weathering rates. The site is estimated by the SWETHRO field workers to be slightly wet, which probably is because of its placement low in the landscape, close to the large lake Vänern. ForSAFE, on the other hand, models soil water content dynamically, using water inputs, texture, and slope along a transect. In this study, we did not have enough data for modelling transects, but settled for modelling a single soil profile for each site. Because of the texture of Södra Averstad, ForSAFE modelled the site as having recurrent dry summers, which seems to be incorrect. The weathering rates in this site might therefore be higher than the model indicates.

The model cannot simulate effects of pests, forest fires, individual trees or whole forest stands dying of natural causes, nor other tree species spontaneously growing in the modelled forest stand and gradually becoming the dominant species. This means that the effect of climate change on the vegetation is underestimated, since for example drought or nutrient imbalances might lead to vulnerability to pests, leading to the tree species at the site being replaced by other tree species, which in its turn

can have large effects on soil chemistry, soil moisture, vegetation nutrient needs and therefor on weathering rates (Augusto et al., 2014). All the modelled sites are planted *Picea abies* stands, but in southern Sweden, it is likely that such stands will not be feasible in the future, especially a future according to the A1B or another high climate change scenario (Grundmann et al., 2011). A future change of forestry practices or tree species might enhance or decrease weathering rates and vulnerability to droughts.

Increasing concentration of atmospheric carbon dioxide leads to a fertilisation effect on the vegetation, which is accounted for in the ForSAFE model and has effects on soil moisture and thus weathering. The combined effects of high temperatures, water stress, carbon dioxide fertilisation, water need, and heat tolerance of the plants are complex (Ruiz-Peréz et al., 2020; Hayatgheibi et al., 2021) and there are signs that the fertilisation effect of carbon dioxide will be lower in a changing climate than what has been previously assumed (Duffy et al., 2021) and that the fertilisation effect already is declining (Wang et al.,
2020). The effect that this could have on weathering rates, base cation availability for vegetation and acidification status of soils, would have to be investigated further.

## 5 Conclusions

The dynamic modelling in this study shows that weathering rates can be substantially affected by climate change, and that the size and direction of the effect varies in time and space. According to the A1B climate change scenario, weathering rates will
increase to 2100 due to higher temperatures, but the increase is distributed differently over seasons. For example, there will be almost no change in winter weathering in northern Sweden, although the temperature change is the highest in winter, since the temperature still will be below zero. Thus, dynamic models like ForSAFE, where seasonal variation is considered, are required for credible assessments of climate change effects on weathering rates and nutrient sustainability.

Weathering rates, and how they are affected by climate change, depends strongly on soil properties. Coarser textures lead to more severe effects of droughts, but also to a more rapid rewetting of the soil during precipitation. This leads to greater fluctuations in weathering rates during drought and rewetting events, implying a greater risk for negative drought effects for forests on coarse textured soils. There is also a difference between regions, where southern sites have more summer drought already today and in the future A1B scenario, with restricting effect on weathering. The difference between sites because of
soil properties are stronger than the climatic effects between regions.

**Author Contributions** Conceptualization, V.K; methodology, V.K., N.S. K.L., G.Z., S.B., C.A.; validation, V.K., K.L., G.Z.; formal analysis, V.K.; data curation, V.K.; writing—original draft preparation, V.K.; writing—review and editing, V.K., C.A, G.Z., K.L., N.S., S.B.; visualization, V.K.; project administration, V.K.; funding acquisition, C.A. All authors have read and
agreed to the submitted version of the manuscript.

**Code/Data availability** Soil, soilwater chemistry, deposition and biomass input data were collected at the Swedish Throughfall Monitoring Network (SWETHRO) and is openly available on the web-site (https://krondroppsnatet.ivl.se/), or on request to IVL, the Swedish Environmental Research Institute or to the authors of the paper. Climate and deposition scenario data were derived from simulations by SMHI (Swedish Meteorological and Hydrological Institute), which is also data host for the data. The code of the ForSAFE model is freely available upon request to the model developer: salim.belyazid@natgeo.su.se. Earlier collaborations with other research groups have shown that new users need an initial period of guidance to be able to independently run the model. Therefore, an initial period of collaboration with the model developers is encouraged with the intent to support new user in the initial stage of their work with the ForSAFE model.

**Conflict of interest** The authors declare no conflict of interest.

## Acknowledgements

The authors want to thank Magnuz Engardt for help with the ECLAIRE input data. We also thank the regional air quality protection associations, county administrative boards and the Swedish Environmental Protection Agency funding SWETHRO and the Swedish Forest Agency (Skogsstyrelsen) for the data provided. This study was partly funded by the Swedish Research Council Formas (reg. no. 212–2011–1691) within the strong research environment Quantifying weathering rates for sustainable forestry (QWARTS), and by The Swedish Environmental Protection Agency, through the programme CLEO (Climate Change and Environmental Objectives).

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

Appendix A

**Observed and modelled soil water ANC in the seven sites**

ANC calculated from measured soil water concentrations are more varied with time, than modelled ANC values during the same time period (Fig. A.1), even though modelled Na and sometimes Cl are more varied with time than the measured values (not shown). At the sites with high sea salt input (especially Västra Torup and Södra Averstad), both Na and Cl have a high variability in the modelled values, but they partly cancel out in the ANC expression. In the model, deposition of all ions varies with the precipitation, with the same concentration in precipitation throughout a whole year, which means that ANC vary less

than it would do if the different ions were deposited more independently of each other and of the precipitation throughout the year.

Only the northernmost site, Ammarnäs, always has ANC values well above zero in both observed and modelled values. The next northernmost site, Högbränna, with its more typical BC poor Swedish forest soil, has a lower ANC, but seems mostly unaffected by acidification both in observed and modelled ANC, both during the period of high atmospheric acidifying

pollution (1950–2000) and in the future. Holmsvattnet, the coastal northern site situated close to large industries, is affected by acidification during the 1980s, with lowered pH and ANC and raised concentrations of aluminium (not shown). According to the model, it is already recovered by 2000. It is then clear cut, temporarily acidifies and recovers again.

All of the four southern sites are to some extent affected by acidification. Observed ANC is sometimes or usually (Västra Torup) below zero. In Västra Torup, measurements from after the clear cut show the large effect a clear cut has on soil water

chemistry, mostly due to leaching of nitrogen. In this application, the model does not manage to reproduce this high nitrogen leaching, leading to lower modelled than observed ANC. Modelled ANC in Södra Averstad show clear acidification in the higher soil layers (not shown), but not at the soil depth where the measurements are made. Except for Bordsjö, the model also shows some recovery of soil water ANC during at least some part of the period 1980–2020. All southern sites except Västra Torup acidify further during the second half of 21$^{st}$ century, even though acidifying deposition is low. This shows that

weathering, even though it increases with the warmer climate, is not high enough to compensate for the forest uptake in the future.

ANC responds to drought with an increased value in the soil water. This increase does not, however lead to export of acid buffering capacity to lower layers or to surface waters, since there is no runoff during droughts.

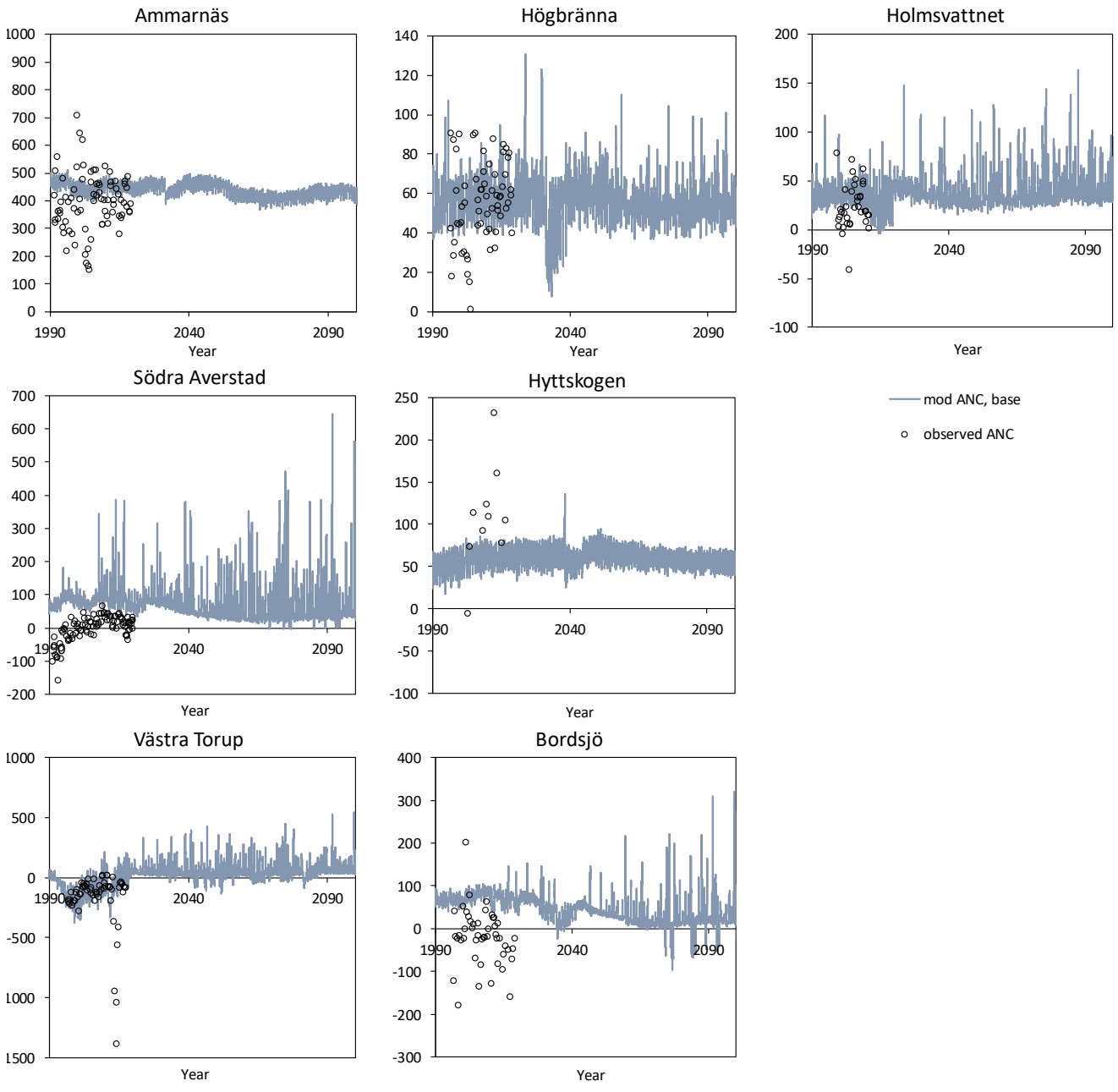


**Figure A.1. Development of soil water chemistry – daily modelled ANC from 1990 to 2100 and observed values three times a year, from lysimeters at 50 cm depth, in the seven sites.**