# Peer review of "Effect of droughts and climate change on future soil weathering rates in Sweden"

_Biogeosciences, 2022_

## Author Comment (AC1)

Reply to comments from anonymous referee #1

Veronika Kronnäs et al.

Author comment on manuscript "Effect of droughts on future weathering rates in Sweden" (Biogeosciences discussion bg-2022-78)

Introduction

We thank the anonymous referee for them taking their time to help us with our manuscript, their impressively thorough reading, critical look and many thoughtful comments that will improve the quality and readability our manuscript. Especially the comments on how to structure the text and present the results in a clearer way will improve the study. We are happy to apply revisions to improve our manuscript as formulated in the answers to the referee comments in this document. We will repeat the comments from anonymous referee #1 in blue, with our answers in black.

*General comments from anonymous referee #1*

*The manuscript "Effect of droughts on future weathering rates in Sweden" by Kronnäs et al. presents model results on how climate change will affect soil weathering across different locations in Sweden. The authors compare the results of different climate change scenarios and identify seasonal weathering dynamics that differ for the various locations. While the general focus of the manuscript is valid and interesting, it is not yet mature enough to be considered for publication. This concerns both the necessity for some language editing, including text organization, and mostly content organization and argumentation.*

*More specifically, text organization is suboptimal, since the information hierarchy is often not considered. It is also not helpful that line numbering starts from 1 with every single page. There are quite a few oversimplifications, e.g. "increasing temperatures increases evapotranspiration…" (line 16, p 2). This is only true as long as there are no drought conditions. Since the manuscript targets Sweden, the general statements in the introduction (and conclusion) should also target Sweden and be backed by references. Some statements appear not-justified, e.g., the shortcut from elevated calcium concentrations in rocks to related higher Ca concentrations in soils. There are various Ca-bearing minerals that do not weather easily. Only with carbonates could such a statement be made. This, more in-depth discussion and argumentation is needed (line 12ff, p 4).*

*Lastly, Figures with multiple graphs of similar content, e.g. figures 3 and 4, should bear the same scales to make a direct comparison easier. Normalizing the absolute weathering rates would make a direct comparison of the relative changes in weathering easier across sites.*

**Authors reply:**

We are happy that you found the subject valid and interesting, and that you put so much effort in suggesting ways of improving it.

We will consider the information hierarchy and hopefully improve the organisation of the text. We will change the line numbers. Regarding the statement on increasing

evapotranspiration, thank you for pointing it out. You are right and we will correct that particular sentence and other oversimplifications. We will also find Swedish references for the introduction.

Regarding the calcium concentrations: We will clarify and not skip steps in the explanation. It is true that there are slowly weathering Ca-minerals, but since we saw elevated Ca-concentrations in soil water, and calcite in other soil samples in papers from the area, we added calcite, not slow weathering Ca-rich minerals or Ca-rich minerals in general.

We agree that the same scale for multiple graphs presented together is preferable, but it was not possible with these values, as they differ so much between sites – something that we also think is an important message in the paper; that weathering rates are so variable geographically, depending on texture and mineralogy as well as climate. We have thought about normalising, but that would also take away information and hide the difference between sites and so we would prefer not to.

*Specific comments:*
*Title: You are not only looking at the effect of droughts, but more general on the effect of climate change. This should probably be addressed in the title and also that you are focusing on soil weathering.*

Agreed, the word soil should be added, and the title should reflect that climate change is included too.

*Abstract: In general, try to focus on your main findings. The abstract contains a lot of descriptive text and it is not always that easy for the reader to extract the main findings. Additionally, try to give some more context, it is not always that clear to what you exactly are referring to (e.g., stronger compared to what, less compared to what…)*

We will try to clarify.

*Line 12, p 1: What do you mean with "climate change base scenario"? Could you give some more context in the abstract already?*

We will do that.

*Line 13, p 1: Either there is a geographical gradient or not. Later you say, that there are clear differences in the response of weathering rates between north and south Sweden.*

There is not a geographical gradient in absolute weathering rates today, but there are geographical differences, especially in how they react to changes in climate such as warming and drought.

*Line 15, p 1: More variable compared to what? Compared to weathering rates during winter or across sites? Try to use less descriptive words and be more precise, e.g. using numbers from your models.*

We will do that.

*Line 18, p 1: "Changes in seasonal dynamics due to climate change differ between regions." This is obvious given the large area your sites cover. In addition, what do you mean with "seasonal dynamics"? Seasonal dynamics of the weathering rate or the climate in general?*

We will be more specific.

*Line 20, p 1: What do you mean with "high"? Please provide some numbers here.*

We will provide numbers.

*Line 24, p 1: Grammar. Use events instead of occurs?*

We will formulate this more understandable.

*Line 25, p 1: Quicker compared to what? Fine-textured soils? Try to be more precise and clearly state where and why weathering is lower/higher compared to what.*

Quicker compared to less coarse soils. We be more precise.

*Line 25, p 1: Weathering rates of 78% still seems to be fairly high to me. It is difficult to put relative numbers in context without knowing the sites and the absolute values.*

We will provide a comparison between weathering rates and e.g. uptake to trees at the site, deposition and/or leaching. This will maybe not fit in the abstract, as it might get too long. In that case the comparison might only be in the results section.

*Line 26, p 1: "In the north, the soils do not dry out as much despite the low precipitation, …" Why? Is it because of less evapotranspiration?*

Yes, it is because less potential and actual evapotranspiration there. We will write this.

*Introduction: As stated earlier, the introduction could be improved by focusing more on Sweden and by mirroring what is presented and discussed in the result, discussion and conclusion sections. In the introduction you focus a lot on the responses of forests, but in the main text you write mostly about soil weathering.*

We will clarify the connection between forest and weathering, write more about weathering and reflect result, discussion and conclusion better

*Line 11, p 2: Better "in contrast" instead of "moreover"?*

The first part of the sentence is about growing base cation (BC) needs because of increased harvest, and the second part is about growing BC needs because of increased growth, so they are not in contrast. We will clarify the writing.

*Line 13, p 2: I cannot really follow your argumentation here. At the beginning of this paragraph you write that forest cover will decrease which will results in a decrease in soil nutrients and here you write that there will be increased forest growth in the future that need more nutrients. Could you please add some more background information and references? In addition, your manuscript focusses*

*more on changes in weathering than the future of forests. So maybe, you should focus more on this and less on the future of forests which are not only driven by weathering (as you also write).*

We will clarify this paragraph and add refences.

In Swedish forestry, harvest is followed by replanting of new forest. Therefore, intensified forestry in the form of more frequent harvest or removal of more of the harvested biomass, does not imply deforestation or changing land use, but rather just more nutrients removed from the soils and the ecosystem. Both faster growth of the trees *and* more intensive forestry leads to more removal of base cations from the soils. If the weathering rates are not large enough to compensate for this removal, the soils might become acidified and/or trees might get deficiencies. To predict if this is likely to happen, modelling the weathering rates is important.

*Line 15f, p 2: "Weathering rates increase with temperature, but also with soil moisture." This is only true if enough moisture is available. Try to be more precise here and explain the importance of the different factors in more detail for Sweden, since this is the geographic focus of your study.*

We will clarify.

*Line 19f, p 2: This is not part of the effect of climate change on weathering. So maybe, this fits better when you talk about trees and how they might response to changes. In addition, to which minerals are you referring here and on which weathering products to they depend?*

It's an indirect effect of climate change, in different direction depending on whether the vegetation grows faster or slower in the new climate. We will clarify the text and which minerals and weathering products are involved.

*Line 20ff, p 2: This is a really general sentence which does not add much information/context. Please give some more context and say under which conditions you expect to have higher weathering rates and under which you expect to have lower weathering rates.*

We will do that.

*Line 22f, p 2: Also, this sentence does not really contain information that help the reader to follow your argumentation.*

We can remove the sentence.

*Line 26, p 2: What do you mean with "this kind"? You have not defined this kind of drought. Please give some context what was so extreme about the drought in Sweden in 2018. How likely is a drought like this in the future?*

We will give more context, describe the drought better (using spei numbers for example) and find information about future scenarios – though there is larger uncertainties in future precipitation than in temperature, and even larger uncertainties in future occurrence of extreme events.

The summer of 2018 was a very dry period in large parts of central and northern Europe. In Sweden, several weather monitoring stations had record low summer precipitation (eg

Lund, which had the lowest summer precipitation of its time series starting in the middle of the 18th century). For most of the country, also the beginning of the year, from February had low to very low precipitation, and despite a more rainy autumn, the yearly precipitation of 2018 was unusually low for almost all of Sweden. The summer of 2018 was also a very warm summer for all of Sweden (1-4 °C warmer summer than the 30-year normal, with the low end only in the mountain chain). (data from SMHI.se, retrieved 2022)

*Line 26, p 2: "nutrient situation" – awkward phrasing*

We will phrase it differently.

*Line 2-5, p 3: This section should be part of the abstract and introduction to define the goal of the study for the reader early on.*

We will remove the subtitle and also include aims in the abstract.

*Line 3, p 3: A1B scenario – you have not explained this abbreviation yet and which climate scenario it is.*

You are right, we have not explained this yet. We will reformulate.

*Methods: This section lacks some more information to fully understand what has been done and why.*

We will go through the methods chapter and include more information, so that it is clear what we have done and why. Though we cannot describe everything about the model or the SWETHRO network methods and they are described in other papers, which we refer to. But we will for example explain the weathering modelling in ForSAFE.

*Line 8, p 3: For how many years have the sites been monitored?*

It is different for different sites, and the exact monitoring years are in table 1. We will write the span of years here.

*Line 9, p 3: What do you mean with "sites' development"? Soil weathering or also other factors?*

Weathering, soil water concentrations, soil moisture, evapotranspiration, runoff, tree growth and other parameters. We will clarify.

*Line 12–22, p 3: The description of the model is really general. What are the assumptions you have to make in order to run the model? What are the input variables? Please provide more context (this could also be part of the supplement). A flow chart could also help here.*

Yes, the description of the model is general, and papers with more thorough description are referenced instead. We will change this and explain the model more, especially the weathering parts in the model and our assumptions better. We cannot describe all input variables and all result parameters, as they are too many.

One assumption of the modelling is that the constituents in the soil water of a site come from deposition and processes in the site, with no lateral transport from upstream soils. This is necessary as we don't have data for any lateral inflow, but it leads to inconsistencies

for the Ammarnäs site, since it must have water flows from soil that is different from the sampled and analysed soil.

Another assumption is that runoff vertically into deeper soil layers beneath the modelled layers always is possible. This assumption is also because of lack of data on when such drainage would be inhibited, but its leads to too low modelled soil moisture at sites that are situated low in the landscape and probably does not always have free drainage downwards, for example Södra Averstad (which is situated close to a large lake in a flat landscape)

*Line 1 – 3, p 4: Missing reference. What do you mean with low weathering rates and "for their ages"?*

We will provide reference. Soils in Scandinavia are relatively young as they were formed at or after the last deglaciation, 16000-9000 years ago (depending on geographical region). Young soils generally have higher weathering rates than million years old soils. But most of Scandinavia have soils formed from granitic or gneissic bedrock (with exceptions at the large islands Öland and Gotland, parts of southernmost Sweden and part of the mountain region, where Ammarnäs is situated), and compared to areas with sedimentary bedrock, these soils have low weathering for their relatively young age.

*Line 7, p 4: Which soil water chemistry variables did you measure and how?*

We will provide more detail and a reference to a paper describing the measurements in detail. Measured parameters are pH, alkalinity, $SO_4^{2-}$, $Cl^-$, $NO_3^-$, $NH_4^+$, TOC, $Ca^{2+}$, $Mg^{2+}$, $Na^+$, $K^+$, Fe, Mn, total aluminium, organic aluminium, inorganic aluminium.

*Line 9, p 4: At which sampling depth? Is your model also depth explicit?*

Sampling depths differs between sites, as it depends on soil horizons. Yes, the soil depths are used in the model. Every soil layer is modelled separately.

*Line 9, p 4: Why do you measure the organic layer? This layer should not be of relevance for soil weathering? Later you write that you use the mineral content of the first mineral layer for the organic layer – this seems to be an oversimplification.*

The organic layer is thin, but very important for other processes than weathering, eg the nitrogen and carbon circulation, so it cannot be omitted from the modelling. The mineral content of the mineral layer in a spruce dominated Swedish forest or plantation is low, though, and the measurement of the total chemistry of the minerals are not trustworthy. As the mineral mass of the organic layer is small, the simplification of using the mineralogical composition of the second layer is good enough. There is no reason to believe that it differs much from the layer right below it, which has the same origin. The amount of weathering from the organic layer will be very small regardless of its mineralogical composition. As an example, BC weathering from the organic layer is around 0.5% of total weathering in the upper 50 cm of the soil profile for Västra Torup.

*Line 9, p 4: What means "most"? Is this variable needed for the model? If so, you should have this measurement for each site. Or explain, why and how you can use the same values across sites.*

We will write more clearly where and how often it has been measured. Standing biomass is not an input variable to the model, as the model models tree growth and standing biomass. But if the measurement is available, it can be used to check the modelled forest growth.

The site Holmsvattnet does not have any measurement of biomass, and the stand has been clear cut, so taking the measurement now is impossible. Södra Averstad, Hyttskogen and Ammarnäs have only one measured value each (which makes calculation of current forest growth between two recent years impossible). The Södra Averstad stand has also been clear cut. Bordsjö has measurements of biomass from 3 occasions and Västra Torup and Högbränna has from 4 occasions each.

*Line 10, p 4: What do you mean by 'few times'?*

1-4 times. We will write that.

*Line 10f, p 4: Can you give some more context? What do you mean it cannot be matched? If the site is so heterogeneous, how can you model it then and how can it be representative for the region?*

It is impossible to have so high concentrations of Ca in the soil water and so high base saturation, if only the low measured Ca content of the mineralogy is influencing the site. Therefore, there needs to exist an influence from some non measured source, such as an lateral inflow of Ca-rich soil water or weathering from easily weatherable Ca-rich minerals in some part of the site where the soil samples were not taken. From other studies we know that there are calcite rich soils in the area.

The region has a steep topography, is very heterogenous in mineralogy, and the site is representative for the areas with higher Ca-concentrations in the soils in this region.

We will formulate the text more clearly, to motivate the choice of site. By including it, we show a larger span of different mineralogies. It is also a site where the ForSAFE model helps us understand the limitations of the measurements we have, by making inconsistencies more visible.

*Line 11, p 4: Better: "… soil water chemistry. This…"?*

Yes, thanks!

*Line 12f, p 4: See my comment in the general section. I do not agree with your argumentation here and maybe you need to investigate this specific site better before you can use it for model exercises.*

Please see the next answer, as well as answers to your other questions about the Ammarnäs soil.

*Line 14, p 4: You already write in the previous sentence that it is likely that there is variability in soil composition. You could easily test this by taking more soil samples, water chemistry and/or rock samples. This should probably be first addressed before modeling your site. Perhaps, your lysimeter and/or soil pit is not representative for the site.*

Taking many more soil samples is unfortunately not easy. It takes time and resources that we don't have in this project. It might be a good next project. But we should not wait with modelling sites until we know everything about them, since the modelling is a tool that

helps us investigate the sites and find inconsistencies and gaps in the measured data. Also, even with these inconsistencies, the site shows the effect of climate and drought in this cold climate region, for example that despite the low precipitation during the drought summers, the soil moisture and weathering will not be as affected as for the other sites – on the contrary, the accompanying warmth even increase the weathering some of the drought summers, compared to the same summer in the base scenario.

That the soil pit is not representative of the whole site, but that the soil in the site is heterogenous, is exactly the explanation that we think is most likely. But the soil samples from the soil pit have very high base saturation, and thus also prove to us that there are some other source of calcium than the weathering from soils exactly like those in the soil pit. The lysimeters are more likely to be representative, as they are more than one and measurements from them have been taken three times per year and always have more or less high calcium concentrations.

We are using data from SWETHRO and part of the basis for our ForSAFE modelling is to use the same type of input data for several sites, with as little extra site specific calibration as is possible. But of course, if we had more soil chemistry data from this site we would have included it.

We could remove the site and have no site in this region, but we would prefer to keep it, since it shows important factors affecting the modelling, this region is relevant for the study and the site is representative of the non homogenous region mineralogy wise.

We will work with the text to clarify it.

*Line 15, p 4: Maybe start a new paragraph after "…soils at the site too.", since you are now writing about the other sites and the land use history.*

Yes, thanks.

*Line 15, p 4: What do you mean with "recently"?*

2011 and 2016. We'll write the years instead.

*Line 16, p 4: What do you mean by "strong"?*

We'll quantify it instead.

*Line 17, p 4: Which elements exactly?*

…and write what elements.

*Line 18, p 4: This is really descriptive and does not help to really understand the land use history of the sites (Why does the reader need to know that the sites were also described in another study?). Give all the necessary information for your study in this manuscript: Why did you selected clear-cutted sites and non-clear cutted sites? How can you compare the different sites in terms of their response to climate change if they have such a different land-use history? Try to provide a more comprehensive description of the sites (focus on the key things that are important for this study) and why you selected these sites - what are you trying to compare among sites? Different climate, land-use etc.?*

We will clarify and remove the reference and the description of the measurements taken after the clear cut in Västra Torup, as they distract.

All of the sites are managed spruce forest sites. In the current Swedish forestry, they will all be clear cut (with the possible exception of Ammarnäs, which is the most likely one to become protected – but it is not currently protected). They all have different planting years and projected clear cut years. This is because there are only about 70 sites in the SWETHRO network, so that sites of all the same age could be chosen. Instead, we chose one of the sites in each geographical region (because that was what we were trying to compare – similar land use in different climate regions of Sweden), that was spruce and had as much of the data needed and wanted (such as tree growth) for modelling as possible. Also, even if two sites have the same planning year, they most likely will not have the same clear cut year, as the clear cut year depends on many different factors (such as the wishes of the forest owner and the productivity of the site), and recommended clear cut age is different in northern and southern Sweden. As we see it, a different clear cut year does not constitute a very different land use history, but is just a detail in the overall land use of managed spruce forest.

*Table 1, p 5: Are the reported values mean or median values? What about standard deviation or the range of the variables? What about detection limits and accuracy? The table is not well integrated in your text.*

The average is the sum of values divided by number of values. Very good idea to add standard deviation to the table, we will do that. And reference the table more in the text.

Accuracy & detection limit: the chemical analyses are performed at an accredited lab, but exact methods, accuracies and detection limits have varied somewhat between the 1980:ies and now. We don't think it is within the scope of the paper to go through that here.

*Figure 1, p 5: This is a very helpful overview figure and should probably come earlier in the text. Why are you providing information about PAR here? You have not discussed it before. How does it influence the weathering rates?*

Thank you! We will move it earlier and explain PAR (photosynthetically active radiation) better. PAR affects the tree growth and therefor indirectly the weathering.

*Line 2, p 7: Which ions are you referring to?*

We will list the ions (which SO4, Cl, NO3, NH4, Ca, Mg, K, Na).

*Line 7, p 7: Please provide this information also in your publication and do not only refer to another publication.*

We will include this.

*Line 8, p 7: What do you mean with "possible mineralogy"?*

We don't measure the mineralogy, we measure only what elements is there. Most of the elements can be part of several different minerals, in different proportions and we don't know which mineral it actually was part of in the soil. We do know, though, that judging

from the measured chemistry, some combination of minerals are impossible (100% apatite for example), whereas other combinations of minerals are possible for this specific soil sample. We will clarify the description.

*Line 8, p 7: What do you mean by "total chemistry"? Did you measure all elements?*

We will explain the concept. We refer to the total amounts of elements of the minerals of the soil, except those lost at high temperatures (so not C, H, O and S). Not all elements are measured, but 20 of the most common: Si, Al, Ca, Fe, K, Mg, Mn, Na, P, Ti, Ba, Be, Co, Cr, Mo, Nb, Ni, Sc, Sr, V, W, Y and Zr. For the A2M-analysis (see next comment) we only use Si, Al, Ca, Fe, K, Mg, Na, P and Ti, as the others are present only in very small concentrations and not used in ForSAFE.

*Line 9, p 7: What kind of model is this? What are the input variables? How does it work?*

We will explain better. A2M is a tool that calculates all possible mathematical solutions to the problem "What can the proportions of different minerals be if the total elemental composition of the soil is X and the composition of the minerals are Y?" (where X is a vector and Y is a matrix). The answer comes as a number (often around 20-30) of different possible mathematical solutions to the question. These can also be combined, so that a linear combination of two solutions is also a possible mineralogy.

*Line 9, p 7: This sentence does not make sense to me. What do you mean by "average mathematical solution"?*

We will reformulate the sentence. The average is the average of all the solutions from A2M.

*Line 10, p 7: As written earlier, why do you need the organic layer for your model? Using the next (mineral layer) does not sound appropriate. Please provide more background to justify your assumption.*

We will explain the model better. ForSAFE is a dynamic model with feedbacks between different processes and stores. It is impossible to use the model with only some of the processes. Omitting a layer that has great importance for some processes would not give good results. Therefore, we need all the input data for the model, even those that might seem less relevant for the specific result we will look at. And the small fraction of minerals in the thin upper organic layer comes from the second layer right below it, so it does make sense to use the same composition of minerals in both these layers.

*Line 11, p 7: Again, how do you know that this is representative for your site, if it does not match with the other measurements?*

Soil water concentrations are measured with five lysimeters at different position at the site, three times per year since 1991, and we think these measurements are representative for the whole site, more so than the soil samples from one soil pit, one time, at the site. The soil pit represent the mineralogical composition of at least part of the site. But even the soil samples from the soil pit show the high calcium amount on the site, in the measured base saturation of 97% (almost all of it calcium). This even further supports our assumption that there must be a larger source of calcium than what the soil mineralogy in the soil pit can provide.

*Line 11, p 7: As an average for what?*

Average over time. We will clarify.

*Line 12, p 7: This seems to be a lot to me. Are you sure this is justified? I think you really need to have a better understanding of this site before modelling it.*

We don't think that 12% is not unreasonably much, as 11-13% has been measured in the area at several sites. The modelling increased our understanding of the site, as modelling often do, but if we find resources for it, it would be very interesting to sample the site and its surroundings, to see if our explanation for where the calcium comes from holds.

*Line 12, p 7: What do you mean by 'has been seen'? Since when can we 'see' mineral content?*

There are optical methods for determining mineralogy, but here we meant measured. We will correct this.

*Line 14, p 7: Awkward phrasing. I would actually argue that all of your sites have a coarse texture and that you cannot really say much about fine-textured soils. To me, your data does not allow you to make assumptions about the effect of soil texture. You could do some simple statistical tests to see if the sites differ significantly in their soil texture (if you do so, you should consider the compositional nature of your soil texture data). In addition, please provide the cut-offs you used to define clay, silt and sand. Furthermore, are the gravel class part of the sand class? Sand, silt and clay should add up to 100%. You can then also use the soil texture classification to say something about the differences (e.g. if there are real differences in soil texture).*

We agree that all our soils have a coarse texture. We are not trying to say anything about fine grained soils and we do point out that all the soils are coarse, though some even more than others. The soils do differ quite much in their water holding capacities because of their different textures (field capacity for one of them is at the level of the wilting point of two of the others, for example), so we do not agree that there is no significant difference between them. We will rephrase to make it clearer that we are not talking about fine grained soils, as we do not usually have them in Swedish forests (but on agricultural land). We will provide the limits between grain size classes. Gravel is a larger size than sand. Sand+silt+clay does not add up to 100% in the input data to the model, since the model also need amount of organic matter as an input value and the missing 2-4 % up to 100% consist of this organic matter.

*Line 16f, p 7: What do you mean with "increasing intervals to the north"? Why?*

Minimum recommended stand age for harvesting is higher in the north of Sweden, because the trees grow slower there. We will clarify.

*Line 21, p 7: Why? Please describe the input variables in brief at the beginning (when you describe the model and what you are exactly trying to model) and explain their role/why they are needed.*

We will improve the description of the model.

*Line 26, p 7: Please describe it also briefly here. It is not good practice just to refer to another publication that did it the same (or almost the same).*

We will do that.

We will, thank you!

*Line 27, p 7: For how many years was it kept constant? What are the time intervals?*

From 1900 to the year before the first year with a measurement of deposition at that site (around year 2000) and from the year after the last year with an available measured deposition to 2100. The deposition is given as yearly values, and ForSAFE makes them into daily, proportionally with precipitation, which is given as a daily value.

*Line 31, p 7: This information is not needed. Maybe delete: "were SMHI was a partner organization."?*

Yes, you are right.

*Line 1, p 8: Are these assumptions you made or are they justified by any measurements or other studies? Please clarify and provide references.*

Dry deposition is very low at an open field and much larger in a mature forest, as supported by SWETHROs measurements. We do not have measurements to support or disprove that it increases exactly linearly between the clear cut forest, which is an open field, and the mature forest, but a linear increase is a more simple assumption than any non linear increase, so we chose this kind of increase. We will provide a reference.

*Line 4–6, p 8: Again, why are these measurements needed and what is their specific role?*

ForSAFE is a dynamic biogeochemical model, that models the forest sites' processes, stores and element flows. Weather data is needed for the processes to be modelled. All processes are modelled regardless of what exact result are wanted – therefor we can't omit any input data because they feels less relevant to the exact aim of our study. We will describe the model better in chapter 2.1.

*Line 11, p 8: What do you mean by bias correction of data from CLEO?*

Bias correction is when data from climate models are adjusted to fit observed local data. We used the method of bias correction in the reference provided (Hempel et al.).

*Line 13, p 8: Is this commonly done? If so, please provide references, if not, explain why you did it this way.*

We assigned climate data from the period 1961-1970 to the years 1900-1960 because we needed climate data for a time period that we need to model but where the exact weather has little influence on the results  in the time periods we actually look at. We need to model the 20th century because the model needs to be run for a time period before the soil properties are measured, to build up the modelled soil stores of the elements. The fact that the climate changes were smaller in this time period than they are now, also make it less

important that we are not using the exact weather of that time period. We will try to find a reference for the method.

*Line 26, p 8: "right level" - Awkward phrasing.*

We will rephrase.

*Line 27, p 8: I am not sure if your approach is better, since you have to make a lot of assumptions and without a more detailed description your exact methods, it is not possible to varify your assumptions and corrections. Why did you make all this effort, if you don't have the data from the sites? Using global products what probably just be fine – as you also write (since your results are similar to earlier approaches).*

We need time series from 1900 to 2100 and we are using data from the European level dataset that the Eclaire program developed, because it has time series that we need, for the sites that we are modelling, although not for the earliest period (which had less of the climate change of the later time period anyways). Maybe our text about how we compared them with local measurements are unnecessary and only confusing.

*Line 31, p 8 to line 21, p 9: This information should probably be moved up. It would help the reader to have this information earlier in the method section to know what you mean with "climate base scenario" and "drought scenario".*

We will move it to earlier in the text.

*Line 14f, p 9: Why are the drought years depending on the year of clear-cutting? I think you really have to make sure that you clearly state what you are modelling, which input variables are need and what are the output variables. I am missing the link between the forest management, and soil weathering and climate change in your manuscript.*

Because really young trees use so much less water that it would not become a severe shortage if the drought years happened to happen right after clear cutting. We will clarify that we were avoiding the period of really small trees, and specify what age we wanted them to be above.

*Line 20, p 9: Why are the numbers in Table 2 are only based on two sites? What about the other sites?*

No, the same relative decrease of precipitation and increase in temperature and radiation are used for all the sites, not just two. We will clarify the text.

*Figure 2, p 10: Since the differences are not that big between the two scenarios, you could try to plot the differences instead of the absolute values for the two scenarios. Similar to your Table 2, but instead of showing % use the actual units. Yet, I am not sure if this will actually improve the figure or not. In your legend you only need to show the color for the two scenarios once; I don't see any squares in the figure, so you could delete this legend.*

We will find a way to improve this figure. Maybe make a table instead, with the differences between the two scenarios.

*Results: The findings are quite interesting, yet, the structure of the text needs to be improved. You may also consider presenting your results in a different value. At the moment, a direct comparison between the sites is difficult, due to their large differences in absolute weathering rates – normalizing the values might help which would allow to directly compare the relative changes across sites.*

Thank you! We will try to structure the results better. The weathering rates are presented as absolute values in the figures and relative in the table.

Line 9, p 11: Are you showing these results somewhere? Without any numbers it is difficult to verify these statements. As I said, your texture gradient is not that big, so I'm not sure if you can actually say that weathering is driven by soil texture. But, I might be wrong, so seeing the data what help here.

It is correct that we haven't tested if the weathering is affected by texture. It is already known that weathering is affected by grain size, since weathering takes place on the surface of the mineral grains and smaller grain sizes have a larger surface area per mass. We will clarify this in the introduction and/or the method section. Our results are not that weathering is affected by texture, but how the texture affects the dynamic of the weathering at droughts and rainfall (that the effect is faster at the coarsest site and less fast in the less coarse sites), and we are also confirming that the combined effect of texture and mineralogy can be stronger than the effect of climate and give a larger weathering in a northern site with a colder climate, than a site far south of it with a much milder climate. We will clarify what we found, what we confirmed and what was known already and built into the model.

*Line 14, p 11: Under which conditions will we see the largest increase in 1990–2019 and 2030–2059, and between 2030–2059 and 2070–2099? Also, didn't you model until 2100?*

We will write out more about what affects when the largest increase in weathering happens for the different sites. Yes, we modelled the year 2100, but it is not part of the 30-year period 2070-2099.

*Line 16, p 11: Awkward phrasing: "In absolute values, Ammarnäs has by far the largest increase in weathering, as it has by far the largest weathering." What are the absolute values and what do you mean by "largest increase in weathering because of largest weathering? This sentence does not make sense to me. Same is true for the first sentence in the next paragraph (line 18, p 11). Maybe you should present your results in a different way, to account for the large differences in the absolute weathering rates which influences your relative values. You could normalize your weathering rates which would make it easier to compare the changes in the weathering rates across sites.*

We will rephrase it. We agree that "largest increase… _as_ it has the largest weathering" is wrong, and we will change it. The values, both absolute weathering rates and relative increases, are given in table 3 and figure 3.

*Line 18ff, p 11: Check sentence structure. If I understand it correctly, you are saying that in southern Sweden there is an increase in soil weathering in all four seasons, with winter showing the smallest increase?*

Yes, that is what we mean. We will check the sentence.

*Table 3, p 12: Add explanation of the acronym BC to your table. A table should stand alone without the text, so all acronyms used in the table should be explained below the table.*

Yes, you are right and we will do that.

*Figure 3, p 13: In the text you refer usually to southern and northern sites. Maybe you can add this information also to your figures to make it easier for the reader to know which site belongs were. You could use a color code for this or describe it briefly in the figure caption.*

Yes, good idea.

*Line 2–13, p 14: Very nice description of the drought scenario and how these changes effect soil weathering in general. It might be useful if you provide some of the information already earlier in the text.*

Thank you! These descriptions of soil moisture and weathering are results from the model, so should be in the results section.

*Line 20f, p 14: Could you provide some more context? Which minerals are causing the differences? This is the first time that you are talking about differences in mineral weathering. You should probably provide some background information for this in the introduction and/or method section.*

Yes, we will add this to the introduction or methods, for example in a description of the weathering modelling in ForSAFE.

*Table 4, p 15: "Averages, with standard deviation around the average in parenthesis." Check sentence structure. Maybe better: Average values with standard deviation in parenthesis. In addition, did you check the data distribution. Median values and median absolute deviation might be better to summarize your data.*

Thank you, that is a better sentence! We want to use averages, as that is what is presented in other papers (since most methods for calculation of weathering rates can calculate average rates only) and what there is to compare our numbers to.

*Figure 4, p 16: Legend is missing and see comment about Figure 3.*

Thank you for noticing! The legend has disappeared in the pdf, and we will put it back.

*Discussion: The discussion contains interesting thoughts, yet the structure of the text is not ideal. Some of the results discussed in the discussion section have not been presented in the result section. Part of the discussion reads like a result and/or method section.*

We have gotten many good suggestions on improving the structure of the discussion and where material should be presented, so we will be able to correct this.

*Line 4, p 17: Again, I don't think you can support your finding that the highest weathering increase will be in summer with the fact that weathering is highest in summer.*

We don't support it on that, we support it on the weathering modelling results. In Table 3 we show the relative weathering increase per season and in Figure 3 we show the absolute

numbers, and the increase is larger both in absolute numbers and relatively during summer, with the exception of Västra Torup in relative numbers.

*Line 9, p 17: What are the implications for plant nutrition? Please support your discussion with references.*

We will clarify and find references.

*Line 13, p 17: Same as above: what are the implications for the ecosystems? Please provide some more background here. I think your discussion goes in the right direction and contains interesting thoughts, yet they need to be supported by references and also stated more clearly with more information.*

Thanks! We will develop this (also taking into account comments on the topic from dr. Katrin Fleischer) and provide references.

*Line 16, p 17: Have you really tested this? I cannot remember reading anything in your result section about this. Is this your own finding or is this statement based on literature? If the latter case is true, provide references.*

We are not sure we understand what part you are referring to, that weathering is affected by soil temperature or that soil temperature is dampened compared to air temperature? We will provide reference – but we can also see it in the model results, and didn't show it in the results as it did not feel like an important result, being rather well known already. But the future change due to climate change in the relation between air and soil temperature would be interesting results to show in the results section, we will see how that can be done.

*Line 16–26, p 17: This should be part of the result section or when you describe the differences between sites. In my opinion, it does not really fit in the discussion here since you have not presented results about soil temperature. This section comes a little bit unexpected and needs to be integrated into the rest of the manuscript. Same is true for the following paragraph about soil acidification. In general, you can only discuss results that are presented in the result section.*

Yes, part of this should be in the results section. Thanks for pointing it out.

*Line 29–32, p 17: This sentence does not really contain information that help the reader to follow your argumentation. Under which condition does soil acidification decreases soil weathering and under which conditions does it increase soil weathering? This could be part of your introduction and in the discussion, you can discuss what you find for Sweden and why it might be different (or not) and what this tells us.*

We will clarify. There are many factors at play at the same time affecting weathering in acidified soils. Some acidified sites have increased concentrations of dissolved Al and some do not, at least not much. According to ForSAFE and the experimental data it is built from, dissolved Al inhibits weathering rates. At the same time, increased H+-concentration enhance weathering, as does increased DOC, which can also be found in acidified soils. In a modelling of Västra Torup (Kronnäs et al., 2019), the acidification of the soil in the 1970:ies inhibited weathering of silicate minerals. The other sites in this paper are not acidified enough for this effect, neither today or in the future up to 2100 according to the scenarios we used, so we don't have this effect and thus cannot discuss in what direction it acted on

these specific sites. But some of them are getting more acidified in the future according to these scenarios, so with a continued forestry as of today or more intense and an ongoing climate change, they might get to an acidified state later.

*Line 26ff, p 18: This reads more like a method section.*

Yes, we should move it.

*Line 4–17, p 19: Not sure what this comparison is supposed to tell me. If understand it correctly, the other models show similar annual values, but do not provide seasonal results. Personally, I don't think you need a separate section for this. You can just point this out, when you present the annual values for your study, saying that these results are similar to other model approaches across Sweden.*

We agree, we will remove this section.

*Line 19, p 19: Accept for Västra Torup, your model does not seem to be capable to capture the variation in ANC. Yet, you have not really discussed the role of ANC for your study.*

Agreed, we do not capture the usually wide variation in ANC well. We think this might be at least partly because of how we give the deposition input to the model, with the different elements having the same distribution as the precipitation during a year (because of the available resolution of the measurements of deposition). In reality there is a larger temporal variability in deposition ANC and chemistry than what we can see in the SWETHRO deposition data, since it is measured with a monthly time step. Nitrate also have a large influence on ANC and for example after a clear cut, the concentration of nitrate is very affected, and the model does not capture the timing of this well at the moment (as can be seen in the measurements from Västra Torup) – but this has little effect on weathering rates, especially outside of that time period, which is short. Therefore, we capture average ANC and temporal trends reasonably well we think (given our knowledge of the sites and the model), but not the individual measured soil water concentrations. The role of ANC in this study is mostly to show if the modelled chemistry is completely off because of some unknown factor (as it was in Ammarnäs before we knew about occurrence of calcite in the area and adjusted the soil mineralogy).

*Line 22ff, p 19: This is an important finding and I think you need to discuss this more. I acknowledge that you have this section about limitations in the main text of the manuscript. Yet, I am wondering if your model structure is appropriate, since it looks like that your model is not capable of capturing site-specific properties.*

There is always a trade-off between complexity and data needs, and the level of complexity in our model is relatively well harmonised with the amount of input data that we can obtain for these sites and what level of complexity we need to investigate the questions we are asking in this study. The model can help us see that something is unexpected going on with e.g. the Ammarnäs site, that there might be a places with more easily weatherable mineralogy on the site or around it than what the soil samples taken there show, but we don't have the means – in this project – to sample the site and its surroundings more thoroughly. Therefore, we would not have the data for a more complex model, with many soil profiles at the same site, in any case. There are also ForSAFE versions that have more than one soil profile, but we cannot use them at these sites because of their larger input data

needs. Except for soil heterogeneity within a site, the ForSAFE version we are using is using site specific data.

*Conclusion: This section summarizes the main findings well. Yet, I am not fully convinced that soil texture is among the main drivers of soil weathering in this study. I would like to see some more statistical tests to back-up this conclusion.*

We know from previous studies that texture is a driver of weathering in this model. In this study, we see (and we will show these results more clearly) that texture also has an influence on how the soils respond to drought and rewetting. We will clarify this.

*Line 23, p 20: "A1B climate change scenario"*

Yes, that is better.

*Line 29, p 20: Again, I think for this conclusion you need to perform a few more tests. Firstly, do the sites significantly differ in their soil texture, secondly, can the results actually be linked to soil texture?*

Yes, the soils differ in texture, with Hyttskogen being very coarse, Västra Torup being less coarse and the rest of them in between, see diagram.

[Figure]

These differences have an effect on water holding capacity, conductivity and weathering rates and we see in the soil water content results from ForSAFE that drying out and rewetting is much faster in Hyttskogen, the coarsest soil, than in the other soils. This is because of the hydraulic properties of the soil due to the texture and it affects the response of the weathering rates to the drought and rewetting. We can add a figure where this is more visible.

*Code/data availability: I highly encourage the authors to publish their data and code in a repository. This can be done independently of publishing the code for the model.*

We will consider this. The model is complex and does not have any user manual and is therefore not really usable without prior knowledge or contact with someone familiar with it.

*Line 13, p 21: The code is not freely available, if it needs to be requested.*

We will consider putting it in a repository.

***References:*** *The doi is not reported in a consistent way and sometimes it is completely missing.*

We will provide the missing doi:s.

---

## Author Comment (AC2)

Reply to comments from Katrin Fleischer

Veronika Kronnäs et al.

Author comment on manuscript "Effect of droughts on future weathering rates in Sweden" (Biogeosciences discussion bg-2022-78)

**Introduction**

We thank dr. Katrin Fleischer for her thorough reading, critical look and many thoughtful comments that will improve the quality and readability our manuscript. Especially the comments on how to structure the text, with many good ideas on how to present the scenarios and the results, which will improve the study. We are happy to apply revisions to improve our manuscript as formulated in the answers to the referee comments in this document. We will repeat the comments from dr. Katrin Fleischer in red, with our answers in black.

**General remarks**

The paper by Kronnäs et al. describes a timely issue of climate interactions with weathering rates under future climate change. Weathering is an important process for nutrient supply and limits plant growth in some ecosystems, and I read the manuscript with great interest. The authors present a model application for Swedish sites along an environmental gradient, which is a suitable study setup to better understand and model weathering dynamically. Current and future rates of weathering are reported for two future scenarios, considering seasonal variation of weathering rates. I appreciate the conclusion that we need dynamic modelling of weathering in space and time, however, I am not fully convinced that the presented study, in its current format, is advancing our understanding of the mechanistic drivers of weathering and how they are affected by climate change. I recommend the authors adopt a more ecosystem-level approach in the analysis and include interactions with plants, and fully explain, analyze, and discuss the underlying mechanisms that lead to the model outcome in terms of weathering.

The paper presents results on seasonal ambient weathering, and effects of a hypothetical drought in future scenarios. The presentation of results lacks important details and needs to be improved by providing clearer visualization items, and correct and clear presentation of quantitative results. E.g. statistics on mean weathering rates, in absolute terms, are not reported in the results section and are not adequately compared to other estimates. The underlying modelled processes in regards to weathering need to be fully explained in the paper, for the reader to interpret the model outcome and better follow the discussion. Also, the underlying soil texture and mineralogy effects are not well explained. Other small details such as units and explanations in captions and headers need to be carefully checked.

While the focus of the paper is clear, the scope of the results seems too narrow, given the presented scientific problem and the model tool ForSAFE. The paper does not present results in its main section on how weathering interacts with nutrient availability from other nutrient-mineralizing processes, nor how seasonal changes in weathering interact with the timing of plants' requirements for soil nutrients. All of which can be modelled by ForSAFE,

and also the introduction leads the reader to believe that these interactions are subject of the study. The scope of the study should be clearly communicated in the introduction and abstract. The paper would improve its scientific depth if these aspects would be addressed within the analysis of the paper. In its current format, results on modelled plant-weathering interactions are reported in the discussion, without visual items or quantifications. Most parts of the discussion are actual material for the results section, and there is little scientific discussion of the model results.

I recommend the authors consider extending the scope of the paper and review the current structure and discussion of the manuscript.

Processes such as nutrient uptake in trees, weathering and decomposition, are interlinked in ForSAFE, which means that interactions with plants are included in the study. We realize that this is not clear from the ForSAFE descriptions in the Methods, and we will therefore add a section where we explain this. We will also add a section about the implications of the interactions on weathering rates in the Discussion.

We realize that the introduction can give the impression that the study will include results on how seasonal changes in weathering relate to the timing of plant nutrient uptake. However, the focus of this study is on weathering dynamics in a future climate, which is something that has not been extensively studied before. This paper is the first one that we are aware of, where weathering dynamics are studied with a daily time step, enabling improved hydrology modelling. We find it important to thoroughly analyse the effects of climate change on weathering rates, before we take the next step, where we plan to compare the weathering dynamics with the plant nutrition requirements in a future climate. We will change the introduction so that the scope of the paper becomes clearer.

In an earlier paper (Kronnäs et al., 2019), we studied the indirect effect of different forestry scenarios on weathering rates, but with a more severe climate scenario (which led to lowered future forest growth), a monthly time step and only two sites in southernmost Sweden. There was a clear effect on weathering of clear cuts, but with very little difference between stem only removal or whole tree harvest. The direct effects of climate change were larger than the indirect effects through feedbacks to plant uptake of base cations. This could be different in different sites and different scenarios, depending on base cation concentrations in the soil water, forest growth rate and other factors.

There are absolute average weathering numbers for each site reported in Table 4, but we could add a table with the averages from the same three time periods that are shown in Figure 3 as well.

We agree on the need to improve the structure of the paper, including the discussion, and we will do this, using the detailed suggestions you and anonymous referee 1 has given us.

**Abstract**

(page 1) L12 Unclear what a climate change base scenario is, and what a drought scenario is.

We will explain this better.

(page 1) L9 Why is the 2018 drought event singled out here? How is it relevant for the study?

It was a recent and unusual summer with very low precipitation for several months in a row, which had a large effect on forests, agriculture, and other sectors of society in Sweden and elsewhere in northern Europe and it is the basis for the drought scenario in this study.

(page 1) L17 Present weathering as absolute rates for better context. Also, in the sentences presenting results afterward.

We will do that, if it does not make the abstract too long.

(page 1) L25 Coarse soils respond quicker in which regard?

In the regard of how fast the weathering decrease during drought occurs, as well as how fast the soil rewets and resume normal weathering rates after the drought.

**Introduction**

(page 2) Methodology of weathering process in the model not well explained. General background knowledge on process-based weathering is not well explained. Too many unreferenced arguments and an unbalanced argument structure.

Agreed, we have not explained the weathering process in the model well enough and we will change this in the description of the model. We will provide references and look at the structure.

(page 2) L8-14 this paragraph is not well balanced in explaining the different drivers for nutrient scarcity in terrestrial ecosystems.

We will look at this text and try to improve it.

(page 2) L15-23 this paragraph is not well structured to follow a common argument. It is unclear what is currently known and where are the unknowns in e.g. how climate change will interact with weathering, and many statements lack an appropriate reference.

Thank you for pointing this out. We will structure the text, find references and be more clear on what is known and what is not.

(page 2) L28 if it is not possible to measure weathering in situ, how is the knowledge derived until now? Explain process knowledge on weathering.

We will explain more.

The processes are known from lab experiments, but it is impossible to let soil sit undisturbed in the lab for as long as it does in nature and therefor the weathering rates are much higher in lab than they are in nature under comparable climate, mineralogy and texture conditions. To assess weathering rates in nature, we need modelling or other indirect methods (like measurements of inflows and outflows to a catchment and calculation of weathering as the unknown to make the other flows add up, or methods that estimate how much of an element that is "missing" from the soil compared to the

unweathered C-horizon, which gives average weathering rate since the soil was deposited at the end of the last glacial).

(page 2) L29 consider rewording here to express actual value of modelling, project into the future, derive hypotheses, etc.

We will develop this sentence to explain this.

**Methods**

(page 3) L3 why use 5 consecutive years, please justify

It was done to see the effect of a more severe drought, as these might get more common in a future warmer climate.

(page 3) Consider removing the small section 1.1 and move aims to introduction.

We will remove the subtitle and integrate the aims better in the rest of the introduction.

(page 3) Consider removing small text after methods and move it to 2.1 and 2.2

We will move it to the end of the introduction, to introduce the methods we use in this study.

(page 3) Section 2.1 Please explain in detail how ForSAFE models the process of weathering, and explain briefly how ForSAFE models other relevant processes that influence the rate of weathering, such as nutrient interactions and plant growth, in order to understand and interpret the modelling outcome.

Yes, we will explain the ForSAFE model better, especially the parts most relevant to the weathering.

(page 3) L18 please specify which research objectives (if you mention that) and which aspects are being improved. This helps understanding the strength and limitations of the model.

We will specify this.

One thing that has been developed is a shorter time step, which makes possible more detailed hydrology modelling, which is useful in this project. One version that can model a set of soil profiles in a row has been developed, which can be used to model the change in soil water chemistry from a hill to a stream.

(page 4) L1 How were the sites chosen, on which data requirement, environmental representation?

They were chosen among the currently or recently active SWETHRO spruce sites that has the required data, and one in each climate region. They are representative of managed spruce forests in their respective region.

(page 4) L2 what does "relatively low weathering rates" mean? Compared to other soils the same age? Can this be more specific?

We will explain better. Soils in Scandinavia are relatively young as they were formed at or after the last deglaciation, 16000-9000 years ago (depending on geographical region). Young soils generally have higher weathering rates than million years old soils. But most of Scandinavia have soils formed from granitic or gneissic bedrock (with exceptions at the large islands Öland and Gotland, parts of southernmost Sweden and part of the mountain region, where Ammarnäs is situated), and compared to areas with sedimentary bedrock, these soils have low weathering for their relatively young age.

(page 6) Why is the A1B scenario chosen? And which climate data is used for the 1990-2019 values? Units of climate variables missing, e.g. mean temperature, annual precipitation sum

The A1B scenario is chosen because it is a intermediate climate change scenario, approximately close to what society today is on track for if no further action is taken, so not unrealistically high, but also not in line with the Paris agreement. Data for 1990-2019 is from the CLEO program. We will be clearer about this.

(page 7) Please explain why the base cation content was not matched, or speculate. It is unclear what the implications of this fix are.

The base cation amount in the soil water, which has been studied by sampling three times per year since 1991, is too high compared to what only deposition and weathering from the measured mineralogy in the soil pit would be able to provide. The exchangeable calcium in the soil from the soil pit was also higher than what these two sources would be able to provide. The reason for this have to be that there are other sources of calcium for soil water and exchangeable calcium on soil particles. These other sources cannot be higher deposition, as too much deposition would be required, but since there are calcite rich soils in the area, inflow of water from calcite rich soils at the site or next to it would be a likely explanation. Therefore, we modelled this.

(page 7) L16 Do you investigate the effect of forest management on weathering rates? Or the effect of vegetation on the modelled weathering rates?

In Kronnäs et al. (2019) we investigated this in detail, but not in this study. These effects are included in this paper as well, since these processes are included in the model, but we do not quantify the effects of different forestry by modelling separate scenarios with different levels of forestry in this paper, as we did in Kronnäs et al. (2019).

(page 8) L1 It would be interesting to quantify the difference in dry deposition due to clear cutting. And also, what is the effect of base cation deposition on forest growth, and how does that interact with weathering in the model simulations?

We will explain the calculation of wet and dry deposition more. The sites are not base cation limited at all, so there is no effect of base cation deposition on forest growth as of now. There are effects on soil acidification or recovery from acidification, runoff chemistry, weathering rates and on base cation content of needles. Often, about half of the input of base cations are from deposition, but the dry deposition is often smaller than the wet deposition.

(page 8) L8 why use 5 consecutive years, please justify

It was done to see the effect of a more severe drought, as these might get more common in a future warmer climate.

(page 9) L2-6 Please show the seasonal variation of precip/T/PAR for the future scenarios.

Yes, good idea, we will include that!

(page 9) L7-9 Due to which factor have they transitioned into another climate zone?

The coldest winter month will be warmer than -3°C as an average, for all sites except the two northernmost, in the future scenario. The second most northern, Högbränna will go from 3 to 5 months with a warmer temperature than 10°C as an average. The northernmost site Ammarnäs will stay in the same climate zone in this future scenario.

(page 9) L11 please define "extreme drought" with quantitative measure, e.g. X reduction in mean precipitation during summer months. I also wonder how often such conditions were projected by the climate models?

Yes, we will give some kind of quantitative measure of the drought. Extreme events are usually not well represented in scenarios from climate models, as they are rare and also the uncertainty in whether they will become more common or not is large.

Figure 2 What is the absolute difference in precipitation between the scenarios? Please quantify the water deficit in meaningful ways, e.g. mean precipitation, monthly MCWD, etc. The difference in temperatures between scenarios is hardly visible on this plot and scale. Also, for precipitation, it would be nice to visualize the summer months, where the actual reduction took place.

We will quantify the water deficit during the drought summers in some way and find a better way to present the data.

**Results**

(page 11) L9 How is weathering dependent on the texture and mineralogy? Please explain before.

Yes, this should be explained in the introduction.

Weathering takes place on the surface of the mineral grains, and smaller grain sizes have a larger total surface relative to the mass, which means that smaller grains have more sites where weathering reactions takes place and thus more weathering. Different minerals have different weathering rates at the same conditions and their weathering reacts differently to changes in temperature, pH, $CO_2$-concentrations, and other factors.

(page 12) Table 3. Instead or additionally, can the response of weathering to warming be plotted? So, the relative control of temperature on weathering can be seen across season, or across mineralogy, etc.?

Good idea, thank you.

(page 12) L15-17 Nice result, this could also be visualized and calculated per site, etc.

Thank you. If this refers to the results on lines 15-17 on page 11, the requested results are presented in table 3.

Figure 3 – why is weathering not increased more in some sites after 2030-59 ? or why is the effect larger or smaller per time frame? Please explain the underlying model drivers and processes.

We will explain the difference at different sites more.

Table 4 – this table is not easily comprehensible. What is the difference in base and drought scenario referring to in %? Is it soil moisture saturation? How long is summer now? It would be better to show % reduction of weathering, in my view. And why show yearly change in weathering, while the drought impacts summer weathering only? Is it possible to depict these results in an informative figure, and perhaps move the data table to the supplement? I am not sure for what kind of patterns I am supposed to look for in the table in its current format.

Thank you for the good ideas. We will change the table. But the weathering is not affected only in summer, since it takes time for the soil moisture to be restored, so for some sites weathering in the warmer beginning of autumn is also clearly affected.

Figure 4 – same as above, it is hard to see differences in weathering, perhaps use a different line size or time-smooth data a little to better visualize.

We will try these suggestions and find some way to improve the figure.

Results section lacks an evaluation of the model's performance in regards to weathering.

The model in itself has been evaluated many times in other projects. In this paper we compare the model results and the soil water chemistry, in the appendix, with the forest growth, and other measurements at the sites, which shows us that our results are as good as we expect them to be for this kind of model on this kind of data (soil water is only measured three times per year, for example, so the measurements themselves do not cover the whole variation in the chemistry). We cannot compare the weathering results with measured data, since there are no measured data on weathering in the field. Our weathering rates are the same order of magnitude that other weathering methods give for these kinds of sites, though (and the differences are partly due to different assumption, soil depths, time periods et c).

Results section lacks reporting of actual modeled weathering rates.

Modelled weathering rates are in figures 3 and 4 and in table 4.

**Discussion**

(page 17) L3-15 This discussion on plant-weathering interactions should be part of the results in my view. The discussion starts with a summary of climate and weathering interactions, and they are now represented more as general remarks and speculations in the

discussion, however, since nutrient supply for forest growth can be modelled with ForSAFE, I wonder why the effect on plants was not further looked into in this study? I see in L. 26 that forest growth is changing dynamically. And in L. 26-29 effects on plant nutrition are actually reported. Please explain how forest growth is treated in ForSAFE in the methods, and consider quantifying nutrient requirements versus nutrient supply via weathering, across the scenarios, sites, etc. I wonder what effect does vegetation have on weathering in your model?

We will explain ForSAFE more thoroughly in the methods section than what we have done now, including the weathering modelling and feedbacks with vegetation. Vegetation has an indirect effect on weathering rates, via its effect on concentrations in the soil water.

Forest growth is modelled dynamically in ForSAFE. The results show that these sites are not growth limited by base cations, neither today or in the future scenario up to 2100, so differences in future growth and contemporary growth in the model results are not due to base cations. If there is an imbalance between needs of base cations for the forest and supple from weathering and deposition (as there is for several of the sites), this first affect soil water chemistry, soil base saturation and runoff acidification, before there are effects on forest growth. There can be other, earlier, effects of suboptimal levels of base cations than decreased growth, such as decreased ability to cope with pests and drought, and ForSAFE does not model these effects, which is why we mention them in the discussion without presenting results. The effect of different forestry on weathering was studied in Kronnäs et al. (2019), but we have not studied the effect on weathering of vegetation vs no vegetation at all. Generally, higher uptake of base cations to vegetation increase weathering.

We will add results, in the result section, showing that the sites are not limited by base cations, and compare the sizes of different fluxes of base cations, from weathering, uptake to vegetation, deposition and leaching.

(page 17) The importance of using dynamic modelling is highlighted only for climate and weathering impacts in this study. The link to plant nutrition is not directly made if I understand the study/model correctly.

In this study, the links to vegetation are in the model (as they always are), but the sites are not growth limited by base cations (as very few Swedish forests are), and early signs on low nutrition on the tree health are unfortunately not included in the model. A next step in another study could be to look at the exact seasonal dynamic of uptake of nutrients and how that compares to stores and weathering amounts during that time period. For this, more detailed measurements of the timing of uptake to trees might be needed and we did not have these for our sites in this study.

(page 17) L20-24 This is rather a result again, the difference in soil temperature and the underlying drivers.

Yes. We will move these results to the results section.

(page 17) L26 consider rewording, "faster … than today in this modelling"

Yes, that is better.

 This is material for the results section. It is necessary to analyze the effect of soil texture on the model results.

We will move these results to the results section and we will show the effect of soil texture on the soil water dynamics and weathering rates during drought and rewetting.

 this is also material for the results section, not the discussion.

Yes. We will move these results to the results section.

 Evaluation of model results need to be done in the results section, and actual measurements and observational-based estimates or model-based estimates need to be included here. A simple statement that they are comparable is not enough.

There are relatively few weathering modelling studies made, and weathering is not possible to measure in the field, so verification against measurements is not possible. Verification against verified observational-based estimates or model-based estimates are not possible, since other modelled weathering rates also are not verified against measurements. The original weathering equations in ForSAFE are verified against laboratory experiments, when the first versions of the model (PROFILE and SAFE) were made. In this study, we refer to other studies to show that our results (averaged over a year or longer periods) are of comparable size, but since these other studies themselves are not verified against measurements (and are modelled or calculated using the exact same assumptions and time periods), maybe this is unnecessary and should be removed.

Instead of evaluating the model based on the weathering rates, we compare soil water chemistry (in the appendix, referred to in the results section) to see if the model results overall are plausible. The ForSAFE model itself is tested in many studies. The modelling of six of these sites are also evaluated in Zanchi et al, which we will clarify in this paper.

 So if weathering has been analyzed with a very comparable approach before, what is the added value of this study? How does this study take us further to what we have previously known?

We will clarify. Weathering was not analysed in that study, which had other objectives, but ForSAFE was used on the same sites except Ammarnäs (as it wasn't relevant for that study).

 now this opens up many questions here, the immediate one is, can you test the effect of CO2 fertilization in your results, simply by keeping CO2 constant in a control scenario. Consider the effect of different CO2 scenarios. That would allow estimating the effect of eCO2 directly. At the very least, the mentioned processes need to be analyzed, e.g. how did soil moisture change or any other driver of weathering due to CO2?

Yes, that is possible and would be a very interesting study. Although it is not the objective of this study, we did model such a scenario as a test. A scenario where the climate changes according to one of the IPCC scenarios, but where the driver of the climate change is missing would be unrealistic, but of value to see what effect every driver has on vegetation on its own and in combination with other drivers. Scenarios with different levels of CO2 could be combined with scenarios with higher or lower nitrogen deposition, since nitrogen often is limiting for growth in Swedish forests in today's climate, and might very well be

limiting the fertilising effect of higher CO2 concentrations also. It would be very interesting to see if ForSAFE how the fertilising effect would be affected by nutrient (mostly N and P in the Swedish context) status and other factors, such as water availability and light. Thank you for the suggestion!

---

## Author Response (AR2)

**Authors response**

Thank you for finding the errors in the manuscript and providing further good suggestions for changes! The paper has improved so much, thanks to all your comments!

**Comments to Co-editor-in-chief Sara Vicca:**

I have changed BC weathering as suggested, even though BC (or Al, Si, Ca et c) weathering is very commonly used in published papers. I have rephrased the rest of the places you suggested too. Thank you for the very useful suggestions! I have tried to improve the language.

**Comments to Anonymous referee #1:**

I have made the suggestions you wanted – thank you for finding my errors! I have not colour coded the sites but instead pointed out in the figure texts which sites are in northern Sweden and which are in southern.

"Line 284ff: Can this assumption (which I think is fine) have an effect on the results. For example, depending on the pre-conditions prior the drought the response might be different?" - If you mean the using of different drought years for different sites: Yes, because the simulated drought is much less severe at the sites if you simulate it at a time when the forest is newly planted, so avoiding this rather short period of young forest does have an effect. The difference in how large the climate change has become in the period 2070-2074 versus in 2090-2094 could also have some effect, but not as much. The sites are situated in different climate regions, with different average temperatures and different local weather, which means that there will be differences regardless of exact drought years, due to for example one site having some drier/warmer/cooler/wetter than average years prior to the drought while another site does not. But the effect of presence of mature trees or small trees on the soil moisture has a big effect and we wanted that important factor to be comparable across sites.

Larger trees use more water – this is clear in the model results, and the developers of the model based this on data, but I don't have the reference right now. I have also seen data on runoff increasing some 200 mm/year after clear cut compared to before clear cut, but I don't have that reference right now either.

"Line 353: I am a little bit confused by the numbers presented here. In Figure 5a, the y axis only goes until ~3%, but the presented values here are much higher. Maybe I am missing something here." – I presume you mean figure 5b, the figure with the relative numbers. In figure 5b, the weathering release rate of BC are release rate of BC during a certain season relative to the yearly average release rate of BC at that site, so that the rate of weathering at summer is larger than 100% of yearly average rate of weathering release, while the winter rate is below 100%. In the text in rows 353, the percentage is percent of increase in weathering release of BC for a certain season and site, compared to the same season and site at the earlier time-period (not compared to yearly average), so it is not normalized to the same number. I changed the axis labels a bit to hopefully make it less confusing.

The effects on weathering from possible changes in $CO_2$ fertilization on trees: these effects would be complex and it is hard to say beforehand in what way they would go. Less water uptake would increase soil moisture, but less uptake of water would also mean less uptake of BC, with perhaps resulting higher concentrations in the soil water, which would lower weathering. Also, less fertilizing effect from $CO_2$ globally would mean less $CO_2$ removed from the atmosphere globally and this would have effects on $CO_2$-concentrations in the atmosphere globally and on climate change, but global feedbacks on climate change is far beyond the reach of the ForSAFE model and would need to be given as input data.

---

## Author Response (AR3)

**Authors response**

I have changed the figure as suggested and corrected the grammatical error in the abstract. Thank you for finding these errors also.